# The membrane-associated proteins FCHo and SGIP are allosteric activators of the AP2 clathrin adaptor complex

Gunther Hollopeter[1,2]*, Jeffrey J Lange[1], Ying Zhang[1], Thien N Vu[2], Mingyu Gu[2†], Michael Ailion[2‡], Eric J Lambie[3], Brian D Slaughter[1], Jay R Unruh[1], Laurence Florens[1], Erik M Jorgensen[2]*

[1]Stowers Institute for Medical Research, Kansas City, United States; [2]Department of Biology, Howard Hughes Medical Institute, University of Utah, Salt Lake City, United States; [3]Department of Cell and Developmental Biology, Ludwig-Maximilians-University, Munich, Germany

**Abstract** The AP2 clathrin adaptor complex links protein cargo to the endocytic machinery but it is unclear how AP2 is activated on the plasma membrane. Here we demonstrate that the membrane-associated proteins FCHo and SGIP1 convert AP2 into an open, active conformation. We screened for *Caenorhabditis elegans* mutants that phenocopy the loss of AP2 subunits and found that AP2 remains inactive in *fcho-1* mutants. A subsequent screen for bypass suppressors of *fcho-1* nulls identified 71 compensatory mutations in all four AP2 subunits. Using a protease-sensitivity assay we show that these mutations restore the open conformation in vivo. The domain of FCHo that induces this rearrangement is not the F-BAR domain or the µ-homology domain, but rather is an uncharacterized 90 amino acid motif, found in both FCHo and SGIP proteins, that directly binds AP2. Thus, these proteins stabilize nascent endocytic pits by exposing membrane and cargo binding sites on AP2.

*For correspondence: guh@ stowers.org (GH); jorgensen@ biology.utah.edu (EMJ)

**Present address:** †Department of Biochemistry, University of Utah, Salt Lake City, United States; ‡Department of Biochemistry, University of Washington, Seattle, United States

**Competing interests:** The authors declare that no competing interests exist.

**Reviewing editor**: Suzanne R Pfeffer, Stanford University, United States

## Introduction

Clathrin-mediated endocytosis is a conserved and ubiquitous process for internalizing material from the cell surface. The Adaptor Protein-2 (AP2) complex serves as a bridge between cargo at the plasma membrane and clathrin. The AP2 core complex contains binding sites for membrane phospholipids and endocytic cargo while the appendages bind clathrin and accessory proteins that coordinate endocytosis (*Traub and Bonifacino, 2013*). AP2 thereby binds target proteins on the surface of the cell and assembles the machinery necessary for internalization of cargo.

AP2 can adopt functionally different conformations. The first crystal structure of the core complex revealed that the binding pockets for cargo and membrane were partially occluded (*Collins et al., 2002*). This structure was proposed to represent a closed, inactive conformation of AP2. Cocrystallization with cargo peptides led to partially open or fully open conformations (*Kelly et al., 2008*; *Jackson et al., 2010*). The open conformation places the cargo- and membrane-binding pockets in coplanar face of the complex and is therefore thought to be the active form of AP2. It has been difficult to determine whether AP2 reorganization is an obligatory process in vivo.

What regulates the switch from closed to open conformation? One model proposes that AP2 can open by simply binding the peptide motifs of cargo proteins and phosphatidylinositol 4,5-bisphosphate ($PIP_2$) on the plasma membrane (*Honing et al., 2005*; *Kelly et al., 2014*). Alternatively, AP2 might require association with clathrin and phosphatidylinositol-3-phosphate to bind cargo at the plasma membrane (*Rapoport et al., 1997*). Another model suggests that the open form of AP2 is

**eLife digest** All cells are enveloped by a plasma membrane. To interact with the outside world, cells constantly recycle the molecules found in, or on, this barrier. This is accomplished by drawing in small patches of the membrane containing these 'cargo' molecules via a process called endocytosis. The predominant method of endocytosis involves coating the tiny membrane pouches with a scaffold-like structure made of clathrin molecules. However, clathrin requires a set of four proteins (known as the adaptor protein-2 complex) to connect the membrane and cargo to the clathrin cage.

Previous studies have suggested that the adaptor protein-2 complex may exist in at least two forms: one in which the binding sites for membrane and cargo are hidden, and another where these sites are exposed. These structures were proposed to represent inactive (closed) and active (open) forms of the complex, respectively. It has been unclear whether reorganization of the adaptor complex is a necessary step in endocytosis or how it might be stimulated.

Now Hollopeter et al. show that worms that lack a membrane-associated protein called FCHo are unable to cluster the adaptor protein-2 complex on their cell membranes, and their cells have difficulties taking up cargo. When the FCHo protein was missing, the adaptor protein-2 complex remained in its closed shape, suggesting that the FCHo protein is needed to switch the complex from its closed to its open structure.

When Hollopeter et al. looked for worms with genetic changes that can overcome the defects caused by a lack of FCHo, they identified worms with various mutations in the genes for the adaptor protein-2 complex. These mutations altered the proteins in the complex at positions that are predicted to rearrange dramatically when the complex is activated; Hollopeter et al. confirmed that such rearrangements do occur in living worms.

Furthermore, Hollopeter et al. found that giving mutant worms, which lacked the *fcho* gene, a small fragment of the FCHo protein causes the adaptor protein-2 complex to adopt its open structure. Similar fragments from other related membrane-associated proteins had the same effect, and these fragments all 'cured' the worms' endocytosis problems. The FCHo fragment directly binds the adaptor complex and Hollopeter et al. propose that FCHo proteins function to activate this complex at the sites where endocytosis occurs.

induced by phosphorylation (*Fingerhut et al., 2001*; *Olusanya et al., 2001*; *Conner and Schmid, 2002*; *Ricotta et al., 2002*; *Honing et al., 2005*). Alternatively it is possible that the complex is activated by one of the many other clathrin-associated proteins.

The most likely of these proteins would be one that precedes AP2 at sites of endocytosis. Examples include Epidermal growth factor receptor substrate 15 (Eps15), intersectin, and most recently, Fer/CIP4 Homology domain only (FCHo) proteins (Syp1p in yeast) (*Stimpson et al., 2009*; *Taylor et al., 2011*). The role of FCHo at endocytic sites is poorly defined. In *syp1* mutants, which encodes the yeast homolog of FCHo, endocytic patches are less frequent, but still progress to coated pits (*Reider et al., 2009*; *Stimpson et al., 2009*). When FCHo proteins were knocked down in tissue culture cells, AP2 failed to bind membrane (*Henne et al., 2010*). However, others found that knockdown of FCHo did not prevent AP2 association with the membrane (*Umasankar et al., 2012*) but that there is an increased tendency for endocytic events to abort (*Cocucci et al., 2012*) with flat clathrin plaques forming rather than clathrin-coated pits (*Mulkearns and Cooper, 2012*). These studies suggest that FCHo might regulate AP2 during the formation of a clathrin-coated pit. On the other hand, there is evidence that FCHo may be acting in a parallel endocytic pathway with ESCRT0 in *Caenorhabditis elegans* (*Mayers et al., 2013*). In fish, FCHo appears to act in BMP signaling during development (*Umasankar et al., 2012*). Thus it is unclear whether FCHo proteins function via AP2 or in parallel to AP2 in clathrin coat assembly, or in an entirely unrelated pathway.

Here, we report that FCHo directly activates AP2 by promoting the open conformation. In FCHo mutants in the nematode *C. elegans*, AP2 is functionally inactive and endocytosis of surface cargo is reduced. However, the requirement for FCHo can be bypassed by mutations in AP2 that specifically destabilize the closed conformation of AP2. FCHo is comprised of an F-BAR domain (*Henne et al., 2007*), a linker region, and a C-terminal µ-homology domain related to the medium subunit of AP2 (*Reider et al., 2009*; *Stimpson et al., 2009*; *Taylor et al., 2011*). The region of FCHo that is required

for activation of AP2 is not the F-BAR or the μ-homology domain but rather a conserved region found in the linker called the AP2 activator domain (APA). This small domain from all metazoan orthologs of FCHo proteins, including SH3-containing GRB2-like protein 3-interacting protein 1 (SGIP1), binds AP2 and is sufficient to activate the AP2 complex in vivo in the absence of the endogenous FCHo protein. We propose that the FCHo/SGIP class of proteins evolved to promote endocytosis by binding to, and stabilizing the open conformation of AP2.

## Results

### FCHo is required for AP2-dependent endocytosis

Mutations in the AP2 complex alpha and mu subunits in *C. elegans* (encoded by the *apa-2* and *apm-2* genes) result in animals with pleiotropic phenotypes including reduced body length (Dpy), egg-laying defects (Egl) and uncoordinated locomotion (Unc). In addition, they exhibit a unique 'jowls' phenotype, in which the mutants exhibit bulges in the cuticle on either side of the head (*Gu et al., 2013*). Deletion of the sigma subunit (*aps-2*) produces a similar 'jowls' phenotype (*Figure 1A* and *Figure 1—figure supplement 1B*), while the beta subunit is shared by both AP1 and AP2 in *C. elegans* and mutations in *apb-1* are lethal. We screened for mutants with the jowls phenotype and identified multiple mutations in three genes coding for alpha adaptin, mu2 adaptin and the nematode homolog of FCHo (*Figure 1—figure supplement 2*). We generated a deletion allele *fcho-1(ox477)* by transposon excision (*Figure 1—figure supplement 1A*); all six mutant alleles of *fcho-1* produced defects strikingly similar to mutants lacking AP2 subunits, including the 'jowls' phenotype (*Figure 1A* and *Figure 1—figure supplement 1B*), suggesting that AP2 function is compromised in the absence of FCHo.

In *C. elegans,* the FCHO-1 protein is localized to the plasma membrane and binds to AP2 in a complex with Eps15 and intersectin (*Mayers et al., 2013*). To determine if FCHo is required to recruit AP2 to the plasma membrane we examined fluorescently-tagged alpha adaptin. In the wild type, AP2 is found in concentrated patches on the plasma membrane (*Figure 1B*). In *fcho-1* mutants, AP2 is associated with the plasma membrane (*Figure 1C*), but does not form clusters, consistent with previous reports (*Henne et al., 2010*; *Cocucci et al., 2012*; *Mayers et al., 2013*). To measure the kinetics of membrane association we performed in vivo Fluorescence Recovery After Photobleaching (FRAP) on coelomocytes. Coelomocytes are scavenger cells which exhibit high levels of endocytosis (*Sato et al., 2014*). The fluorescence signal recovered approximately three times faster after bleaching in the absence of FCHo (*Figure 1C,D*). Thus, FCHo stabilizes patches of AP2 on the membrane and limits its mobility, consistent with previous studies (*Henne et al., 2010*; *Cocucci et al., 2012*).

To determine whether clathrin-mediated endocytosis is compromised in *fcho-1* mutants, we assayed endocytosis of a fluorescently tagged transmembrane protein. This molecule is comprised of a GFP-tagged CD4 protein with a tyrosine cargo recognition motif (*Figure1—figure supplement 1C*). Tyrosine motifs comprised of YxxΦ, where x is any amino acid, and Φ is a large hydrophobic residue, bind the mu2 subunit of AP2 and are required for AP2-mediated internalization (*Ohno et al., 1995*; *Owen and Evans, 1998*). In wild-type worms, very little CD4-GFP is expressed on the surface of intestinal cells (*Figure 1E*). However, this cargo accumulates on the cell surface in mutants lacking AP2 subunits and in *fcho-1* mutants, suggesting that AP2-dependent endocytosis is defective in *fcho-1* mutants (*Figure 1—figure supplement 1D*). In addition the phenotype is not enhanced in double mutants indicating that FCHO-1 acts in the same pathway as AP2.

### Mutations in AP2 bypass the requirement for FCHo

To identify components downstream of FCHo, we performed a genetic screen for mutations that suppress a null mutation in *fcho-1*. To increase the probability of getting missense mutations we used the mutagen *N*-ethyl-*N*-nitrosourea (ENU), which can generate transversions and can therefore swap charges, or hydrophilic and hydrophobic amino acids. In addition, we designed a multigenerational screen to select for subtle improvements in fitness. Wild-type animals grow rapidly and starve a culture plate in 5 days, whereas *fcho-1* mutants exhibit reduced fecundity (*Figure1—figure supplement 1E*) and require twice as long to consume the same amount of food (*Figure 1—figure supplement 1F*). We selected for suppressors that rapidly starved plates, and identified 71 dominant mutations that confer increased fitness to *fcho-1* mutants and suppressed the jowls phenotype. All of these suppressed strains contained second site missense mutations in one of the four subunits of AP2 (*Figure 2—figure supplement 2*) and none exhibited loss-of-function phenotypes for these adaptin

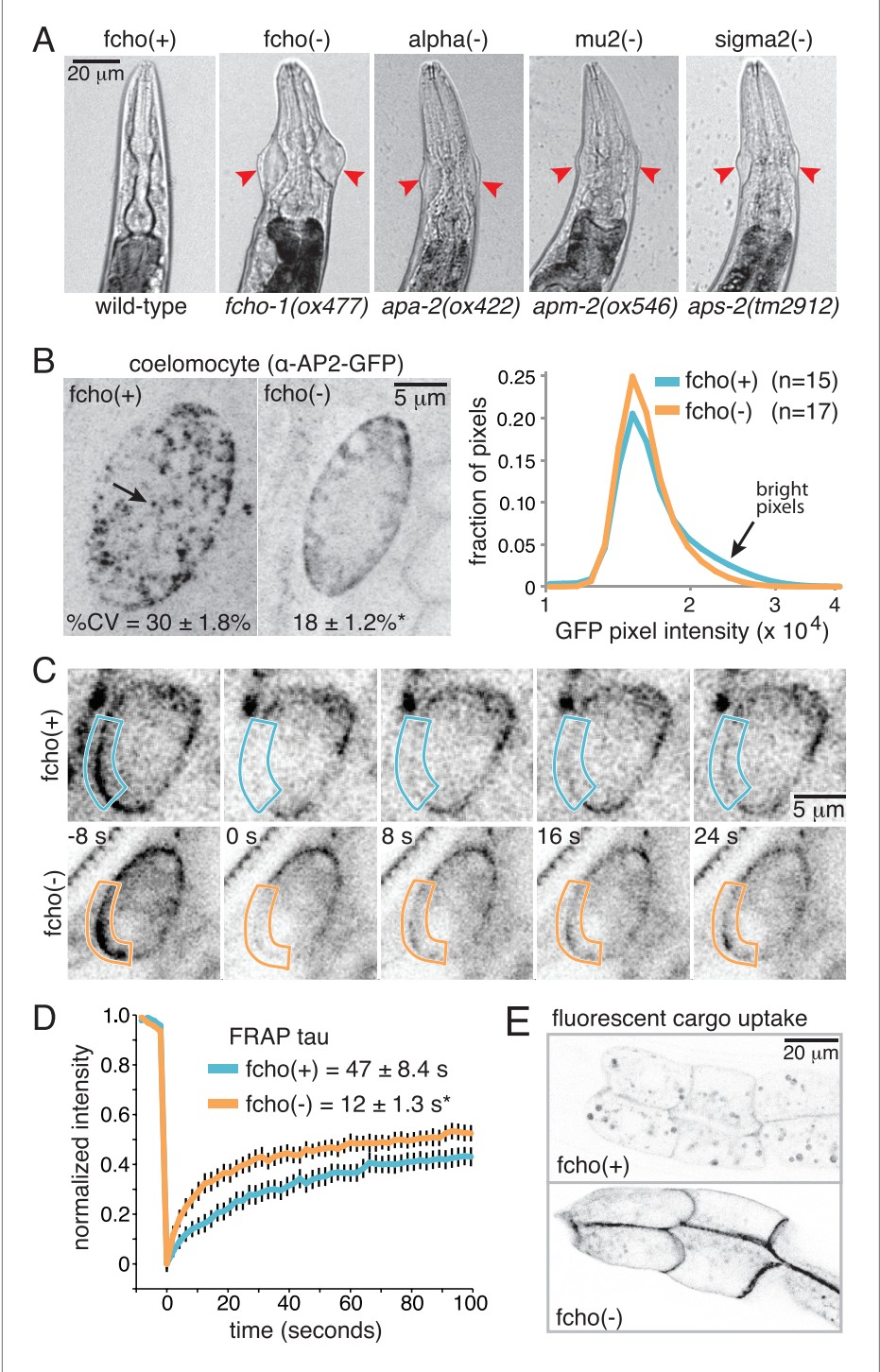

**Figure 1**. Loss of FCHo compromises AP2 activity. (**A**) Animals cropped to highlight jowls (red arrrowheads, anterior up) shared by *fcho-1* and AP2 subunit mutants (*apa-2*, *apm-2*, *aps-2*). (**B**) Left, representative confocal micrographs of coelomocytes in worms expressing GFP-tagged alpha subunit. Images represent maximum projections of Z-slices through ~1/2 of a coelomocyte. Numbers indicate the coefficient of variance of pixel intensities across coelomocytes (excluding the cell periphery). *p < 0.01 unpaired, two-tailed t-test. Right, normalized histograms of pixel intensities (logarithmic scale). Arrow indicates higher intensity pixels that are missing in *fcho-1* mutants. (**C**) Time-lapse montages of FRAP experiments on coelomocytes expressing alpha:GFP in adult hermaphrodites. The outlined membrane region was photobleached at time = 0. (**D**) FRAP assay. Average recovery curves and time

*Figure 1. Continued on next page*

*Figure 1. Continued*

constants of fluorescence after photobleaching. *p < 0.01 unpaired, two-tailed t-test on data from 12 fcho(+) coelomocytes and 14 fcho(−) coelomocytes. (**E**) Cargo assay. Micrographs of intestinal cells (anterior left) expressing a GFP-tagged transmembrane cargo internalized by AP2. The cargo is a truncated CD4 transmembrane construct with a YxxΦ cargo recognition motif (*Figure 1—figure supplement 1C*). The average pixel intensity along an intestinal basal-lateral membrane in fcho(+) animals (n = 11) is 972 ± 85 arbitrary units (au) and 5610 ± 416 au in fcho(−) mutants (n = 12). p < 0.01 unpaired, two-tailed t-test. Data in (**B**), (**D**) and (**E**) represent the mean ± SEM.

The following figure supplements are available for figure 1:

**Figure supplement 1**. FCHO-1 and AP2 regulate the same pathways.

**Figure supplement 2**. Recessive alleles isolated from genetic screen for 'jowls' phenotype * independently identified 'jowls' mutant.

genes (*Figure 2—figure supplement 1A–C*). These mutations all occur at conserved amino acids, and cluster at sites likely to stabilize the closed (inactive) conformation when placed on the crystal structures of AP2 (*Figure 2A–C*) (*Collins et al., 2002*; *Kelly et al., 2008*; *Jackson et al., 2010*). These mutations can be classified into four groups: (1) residues that lie in the bowl-like interface between the mu2 subunit and the other three subunits, (2) residues that stabilize the insertion of the N-terminus of the beta subunit into the cargo binding motif of sigma, (3) residues in the alpha subunit that are found in the helical solenoid that lies across the top of the complex, and (4) the phosphorylation site on the mu2 subunit. It is likely that these mutations destabilize the closed conformation of AP2, suggesting that the open conformation of AP2 may bypass the requirement for FCHo. In other words, these mutations would promote an open conformation of AP2, suggesting that AP2 may dwell in the closed state in the absence of FCHo.

## *fcho-1* suppressor mutations promote protease-sensitive, open AP2

To determine if AP2 remains in the closed conformation in *fcho-1* mutants we devised an in vivo protease assay. The mu2 subunit of AP2 becomes sensitive to trypsin when the complex is incorporated into clathrin coats (*Matsui and Kirchhausen, 1990*; *Aguilar et al., 1997*). The protease-sensitive segment is contained within ~15 residues that are not resolved in crystal structures (*Owen and Evans, 1998*; *Heldwein et al., 2004*), but the boundaries of this segment are apposed to the internal face of the sigma subunit in the closed structure, and are exposed on the exterior of the complex in the open structure (*Figure 3A*). Because this poorly conserved region tolerates the insertion of various tags (*Nesterov et al., 1999*; *Jackson et al., 2010*), we inserted a TEV cleavage site in this sequence, and replaced the endogenous *apm-2* gene with the protease-sensitive version of mu2 tagged with HA (*Figure 3—figure supplement 1A*). We used a temperature-inducible promoter to transiently drive expression of TEV protease in these transgenic worms. After heat-shock induction of the protease, the level of the full-length subunit declined and a smaller 25kd N-terminal fragment accumulated over a 8 hr period (*Figure 3B,C* and *Figure 3—figure supplement 1B,C*) The mu2 cleavage rate was slower in *fcho-1* mutants even though the protease was induced to a similar level (*Figure 3B,C* and *Figure 3—figure supplement 1B-D*). TEV-sensitivity was also demonstrated for a FLAG-tagged version of mu2, which was used in the structure-function experiments described below (*Figure 3—figure supplement 1*). These data suggest that a larger fraction of AP2 is in the closed, protease-resistant state in *fcho-1* mutants.

We also examined the conformation of AP2 in the strains carrying the *fcho-1* suppressors using the TEV protease assay (*Figure 4A,C*). We tested two suppressors each for the alpha, beta and mu2 subunits, and each led to an increase in mu2 cleavage in the double mutants compared to fcho(−) alone (*Figure 4C*). These data suggest that AP2 is in the closed conformation in the absence of FCHo and that single amino acid changes in the complex are sufficient to tip the equilibrium toward the active conformation. Indeed, the amount of rescue observed in the cargo assay (*Figure 4B*) was grossly correlated with protease sensitivity. However, none of these mutations fully restored cargo endocytosis, even though all of the suppressors rescued growth and morphology (*Figure 4A* and not shown). Only the mutation that resulted in a profoundly protease–hypersensitive complex (µE306K) increased cargo internalization with high significance in *fcho-1* mutants. These findings indicate that

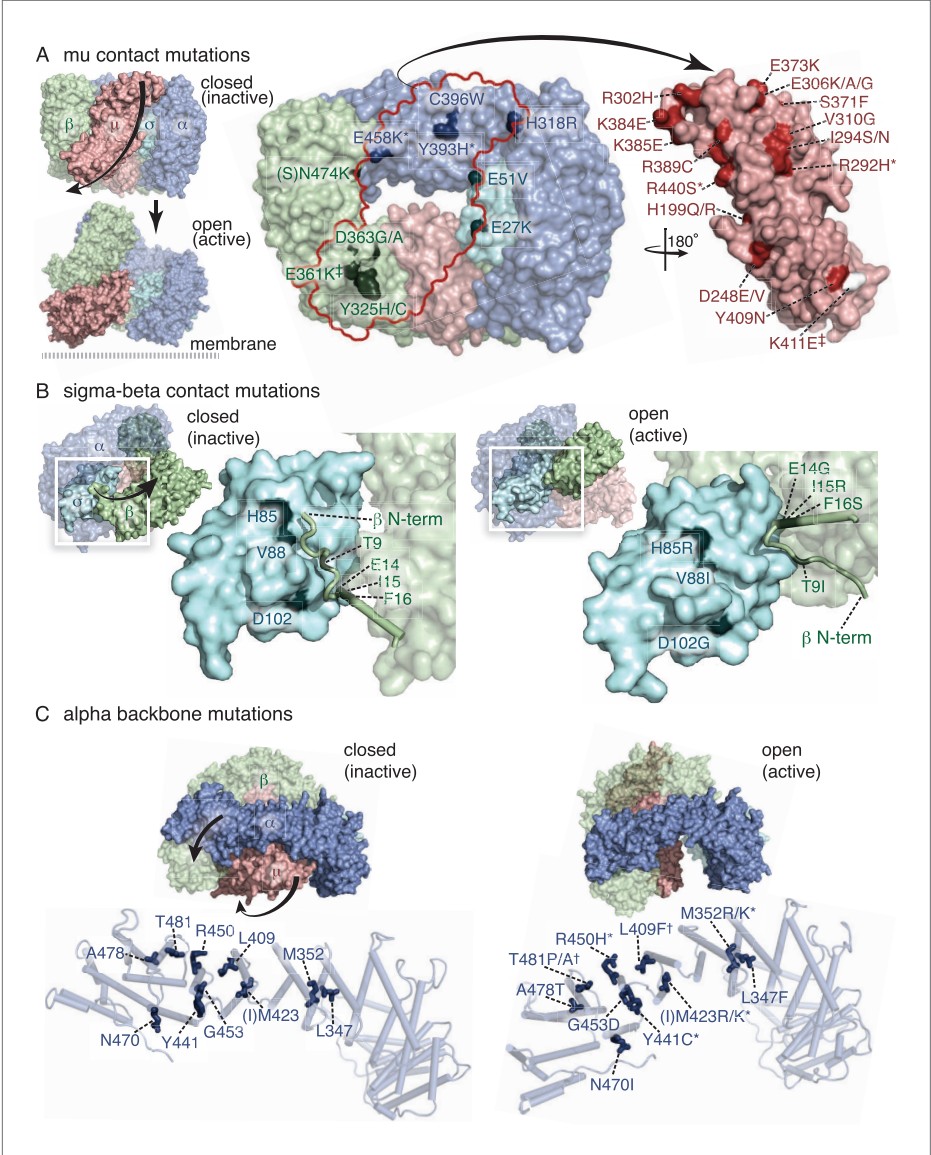

**Figure 2**. Mutations in AP2 closed conformation interfaces suppress fcho(−). Predicted location of the mutated worm residues within the inactive (PBD ID: 2VGL) and active (PBD ID: 2XA7) crystal structures of the vertebrate AP2 core complex. Alpha is blue, beta is green, mu2 is pink, and sigma2 is cyan. The residue numbers are from the worm subunits and parentheses indicate the corresponding vertebrate residue. * designates mutations isolated twice, and † designates mutations isolated thrice. ‡ designates mutations that were combined to re-establish a salt bridge between beta and mu (See *Figure 5*). (**A**) Mutations at the contact interface between the mu domain and the other three subunits. These contacts are disrupted upon opening. To visualize the contact surface in the closed conformation, the mu domain has been flipped to the right. Small renderings (left) show the closed (*Collins et al., 2002*) and open conformations (*Jackson et al., 2010*); the plasma membrane would be below the complex in this view. The K411E mutation on the mu domain (white residue) was not isolated from the fcho-1 suppressor screen, but was engineered (See *Figure 5*). (**B**) Mutations in the latching mechanism formed by the N-terminus of beta and the di-leucine motif binding-pocket of sigma2. (**C**) Mutations in alpha cluster along the hinge region that flexes during opening.

The following figure supplements are available for figure 2:

**Figure supplement 1**. Suppression of fcho-1 by missense mutations in individual AP2 subunits.

**Figure supplement 2**. Dominant mutations in AP2 subunits identified in fcho-1 suppressor screen.

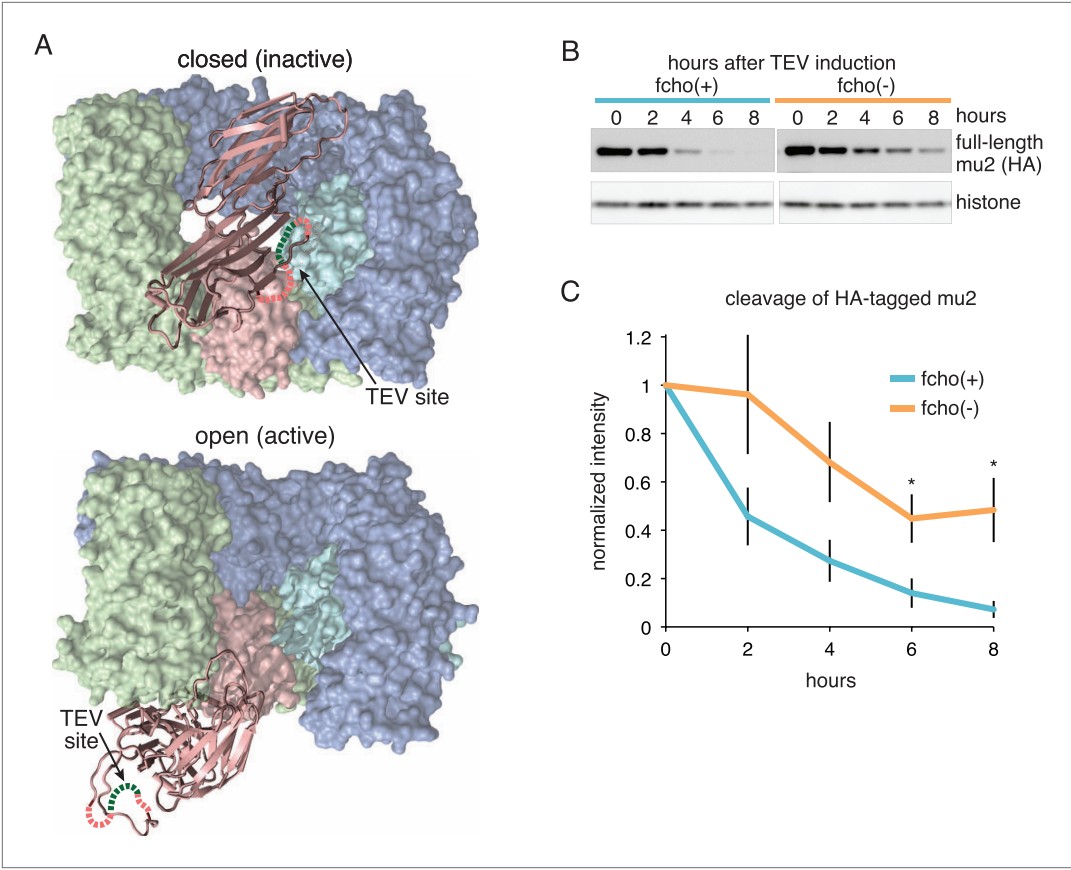

**Figure 3**. FCHo promotes the protease-sensitive open conformation of AP2 in vivo. (**A**) A TEV protease site was inserted into a surface loop of the mu domain. The dashed line connects the boundaries of the unstructured region within two conformations of the AP2 complex. (**B**) Western blot of whole animal lysates expressing the HA-tagged mu2 subunit depicted in *Figure 3—figure supplement 1A*. The amount of full-length subunit (top) decreases following heatshock. Anti-histone blot is below. Each sample is comprised of 100 larval 4 stage animals. (**C**) Quantification of mu2 proteolysis. Intensity of anti-HA signal relative to histone, normalized to time 0. *p < 0.05, unpaired, two-tailed t-test compared to fcho(+) values at same time point, n = 4. Data represent the mean ± SEM. See *Figure 3—figure supplement 1* for results using FLAG-tagged version of mu2.
The following figure supplement is available for figure 3:

**Figure supplement 1**. Schematic of TEV Protease assay and results from FLAG-tagged version of mu2.

subtle conformational changes favoring active AP2 satisfied an organismal requirement for FCHO-1 without fully compensating for the endocytic defect of *fcho-1* mutants.

## Phosphorylation of AP2 is downstream of *fcho-1* function

The vertebrate mu2 subunit is phosphorylated in a clathrin-dependent manner at threonine-156 by Adaptor-Associated Kinase (AAK1) (*Pauloin and Thurieau, 1993*; *Conner and Schmid, 2002*; *Conner et al., 2003*; *Jackson et al., 2003*). The phosphorylated core complex binds cargo motifs and phospho-inositides with higher affinity (*Honing et al., 2005*). In the *fcho-1* suppressor screen we isolated multiple mutations in the equivalent residue (T160) of the worm mu2 subunit, including a mutation to the phosphorylation-defective amino acid alanine (*Figure 4A*, *Figure 2—figure supplement 2D*). Mutating this residue to the phosphomimetic residue glutamate also suppressed *fcho-1* mutants (*Figure 4A*). Note that cargo internalization was compromised in both the phosphorylation-defective and phosphomimetic mutants (*Figure 4B*), which is consistent with previous reports (*Olusanya et al., 2001*; *Semerdjieva et al., 2008*; *Jackson et al., 2010*). Our data suggest that the presence of a threonine at this position stabilizes the inactive state and that any change to this residue is likely to destabilize the inactive state of AP2.

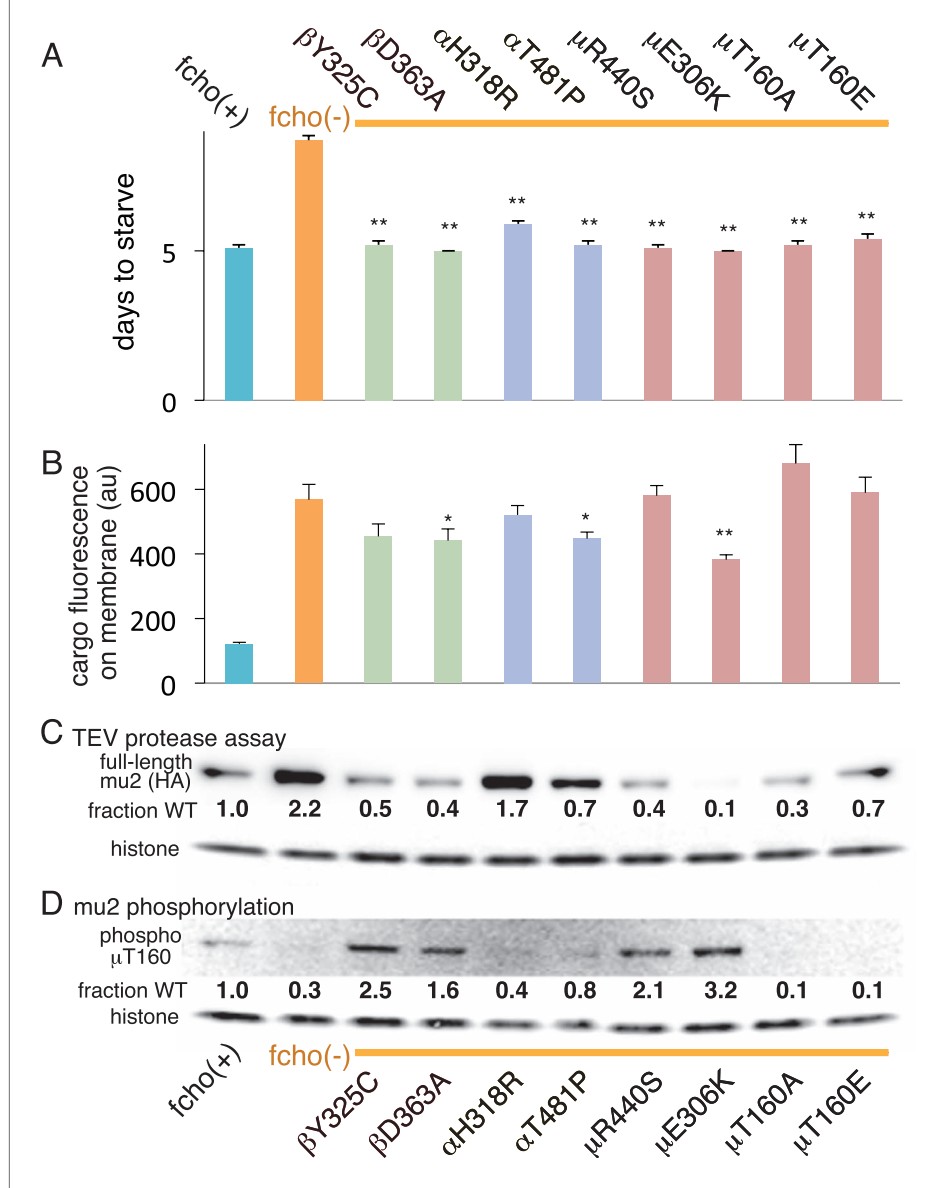

**Figure 4**. AP2 mutations restore the active conformation in *fcho-1* mutants. Listed mutations correspond to the worm residues. (**A**) Starvation assay (days required for a worm population to expand and consume the bacterial food). (**B**) Cargo assay (amount of GFP-tagged cargo on intestinal cell membrane). For (**A**) and (**B**), data represent the mean ± SEM for n ≥ 10. Significance determined by unpaired, two-tailed t-test compared to fcho(−), *p < 0.05 and **p < 0.01. (**C**) in vivo TEV protease assay. Samples collected for Western blot analysis 8 hr after heatshock (as in *Figure 3B*). Numbers indicate band intensity of full-length mu2 (anti-HA, top) relative to histone (bottom), normalized to the fcho(+) sample. (**D**) Blot for phosphorylated threonine-160 (T156 in vertebrates) in the linker region of the mu2 subunit. Samples collected before heatshock. Numbers indicate band intensity of phosphorylated T160 (top) relative to histone (bottom), normalized to fcho(+) sample. For (**C**) and (**D**), each sample is comprised of 100 larval 4 stage animals.

The following figure supplement is available for figure 4:

**Figure supplement 1**. Activation of AP2 strengthens membrane association, enables cargo binding, and stabilizes the location of mu2 phosphorylation.

To assay phosphorylation of the mu2 subunit, we used an antibody specific to phosphorylated T160. We found that AP2 is phosphorylated in the wild-type, and is hypo-phosphorylated in the *fcho-1* mutant (*Figure 4D*). All of the suppressor mutations we tested increased phosphorylation relative to

the *fcho-1* mutants (except of course the T160 mutations themselves). Increased phosphorylation was also associated with increased protease sensitivity of the mu2 subunit in the TEV assay (*Figure 4C*). These data demonstrate that the open state is phosphorylated, and that FCHO-1 is not absolutely required for phosphorylation, but rather AP2 in the active state is sufficient to induce phosphorylation. Nevertheless, this threonine residue is not completely exposed in the crystal structure of the open conformation (*Figure 4—figure supplement 1*) (*Jackson et al., 2010*) so it is unclear whether the side chain would be accessible to AAK1 in this state.

## A compensatory salt bridge mutation restores *fcho-1* mutant phenotypes

The protease sensitivity of the suppressor mutations indicates that the closed structure determined by X-ray crystallography is an authentic structure in vivo, and that these mutations destabilize the closed state of AP2. Nevertheless, it is possible that the mapping of these mutations onto the crystal structure is coincidental. To verify that the closed structure has in vivo significance we identified a mutation among our suppressors that would disrupt a salt bridge in the closed conformation, and used the crystal structure to predict a compensatory mutation that would restore the salt bridge. In the closed conformation β (E361) forms a salt bridge to μ (K411) (*Figure 2A*; *Figure 5A*; *Figure 5—figure supplement 1*). We therefore analyzed mutations in β (E361K) and μ (K411E) that break this salt bridge, and found that both suppressed the *fcho-1* mutant phenotype (*Figure 5B*). These mutations also increased protease sensitivity relative to the *fcho-1* mutant, and increased phosphorylation of threonine-160 (*Figure 5D,E*). Similar to previous results (*Figure 4*), only the mutation that produced an acutely open complex (μK411E) significantly rescued the cargo-recycling defect of *fcho-1* mutants (*Figure 5C*). We then constructed the double mutant containing both the βE361K and μK411E mutations which should restore the salt bridge. The two mutations together no longer suppressed the *fcho-1* growth phenotype or cargo retrieval defect of the *fcho-1* mutants, and reversed the protease sensitivity of the single mutants. Phosphorylation of T160 in the double mutant was reduced relative to the μK411E single mutant but was not fully restored to fcho(−) levels. These results confirm that the closed form as determined by crystallography predominates in the *fcho-1* mutant and that destabilizing the closed form can bypass the requirement for FCHO-1.

## A conserved segment in the FCHo linker domain opens AP2

Which domain of FCHo activates AP2? FCHO-1 is composed of a membrane-binding F-BAR domain (*Henne et al., 2007*), a linker region, and a C-terminal μ-homology domain related to the medium subunit of AP2 (*Reider et al., 2009*) (*Figure 6A*). We generated single copy transgenic animals expressing proteins deleting each of these domains in the *fcho-1* null background. We found that the N-terminal F-BAR domain is dispensable for rescue of all *fcho-1* phenotypes (*Figure 6B* and *Figure 6—figure supplement 1A-D*). Constructs lacking the C-terminal μ-homology domain (μHD) failed to rescue cargo endocytosis, but increased the growth rate, protease sensitivity and phosphorylation of the mutants (*Figure 6B*). However, deletions that extend into the linker domain failed to rescue *fcho-1* mutants. The linker domain was previously found to bind the AP2 complex using pull-down assays (*Umasankar et al., 2012*), suggesting that the activation of AP2 by FCHO-1 observed here could be via a direct interaction.

The linker domain of FCHO-1 is poorly conserved in general, but there is a small region of ~90 amino acids that is shared with other FCHo homologs and the vertebrate protein SGIP1 (*Figure 6A*) (*Reider et al., 2009*; *Uezu et al., 2011*). Expression of this short fragment alone was capable of rescuing *fcho-1* mutants, including growth rate, endocytosis of cargo, and morphology (*Figure 6C*, *Figure 7—figure supplement 1A*, and not shown). This fragment is also sufficient to immobilize AP2 on the membrane in the photobleaching assay and to cluster AP2 into presumptive endocytic pits (*Figure 6—figure supplement 1*). Moreover, the equivalent domains from mouse FCHo2, mouse SGIP1, and human FCHo1 also rescued *fcho-1* mutants (*Figure 6C*), though the fragment from mouse FCHo1 rescued poorly (*Figure 7—figure supplement 1A*). These results suggest that this small 90 amino acid region, called the AP2 Activator motif (APA), encompasses a large fraction of FCHO-1 function in vivo.

To determine whether the APA domain binds AP2, we expressed the worm and mammalian APA domains fused to a HaloTag in tissue culture cells (HEK293) and pulled down the APA fragment using chloroalkane beads. Silver-stained gels (*Figure 7A*) and Western blot analysis (*Figure 7—figure supplement 1B*) suggested the presence of all four AP2 subunits in the pulldowns. To systematically

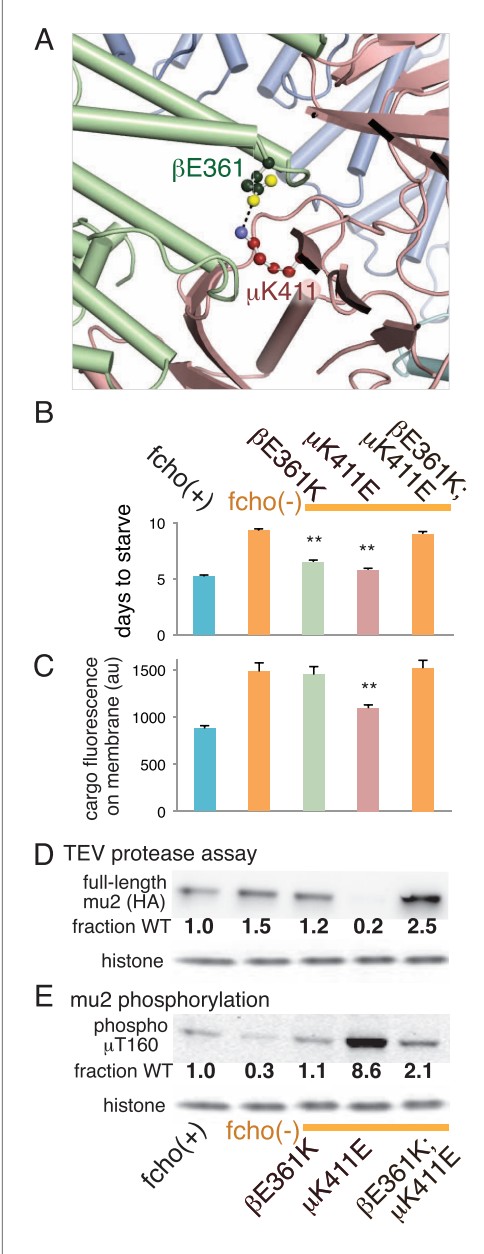

**Figure 5**. Charge swaps activate and inactivate AP2 in vivo. (**A**) Predicted location of residues stabilizing an important inter-subunit salt bridge within the inactive (PBD ID: 2VGL) crystal structure of the vertebrate AP2 core complex. Alpha is blue, beta is green, and mu2 is pink. The residue numbers are from the worm subunits. See **Figure 2A** and **Figure 5—figure supplement 1** for localization on interfaces. (**B**) Starvation assay (days required for a worm population to expand and consume the bacterial food). AP2 mutations indicated above. (**C**) Cargo assay (amount of GFP-tagged cargo on intestinal cell membrane). For (**B**) and (**C**), data represent the mean ± SEM for n ≥ 10. Significance determined by unpaired, two-tailed t-test, **p < 0.01. (**D**) in vivo TEV protease assay. Samples collected for

identify the binding partners we performed a mass spectrometry analysis of the pulldowns using Multi-Dimensional Protein Identification Technology (MudPIT). The majority of the peptide reads were from the bait itself (**Figure 7B**), but the most abundant interacting partner for the APA domains from mouse FCHo2 and SGIP1 was the AP2 complex (~10% of peptides, **Figure 7B** and **Figure 7—figure supplement 2**). When the bait included the entire FCHo2 or SGIP1 proteins (or FCHo2 without the BAR domain), AP2 was still enriched; but additional components known to bind the μ-homology domain, such as Eps15, were also isolated (**Figure 7—figure supplement 1C**). The interaction of the APA domain with AP2 likely occurs in vivo as well, since fluorescently tagged APA colocalizes with AP2 on the membrane in coelomocytes, and membrane association is lost in mutants lacking the mu2 subunit (**Figure 7—figure supplement 1D**).

We demonstrated that the interaction between the APA domain and the AP2 core is direct using purified recombinant proteins in pulldown assays (**Figure 7C**). The APA domain does not appear to bind the appendages of the large adaptins, nor to the mu domain alone. Rather, it bridges the complex since the APA bait binds both the alpha/sigma and beta/mu hemicomplexes. Together these data suggest that the APA domain in FCHo homologs from worms, mice and humans binds AP2 to destabilize the closed conformation and promote the active conformation.

## Discussion

Mutations in the FCHo gene were found to phenocopy loss of AP2 subunits in the nematode *C. elegans*, suggesting these proteins act in a single pathway. We found that FCHo acts upstream of AP2: the function of FCHo can be bypassed by constitutively open mutations in the AP2 complex. Moreover the effect of FCHo on AP2 is direct: a 90 amino acid fragment of the linker domain in the FCHo family proteins binds AP2. This activator fragment is necessary and sufficient for rescue when overexpressed in *fcho-1* mutants.

### FCHo acts on AP2

There is broad agreement that FCHo acts early and it acts to stabilize nascent clathrin-coated pits via AP2. The AP2 complex associates with the plasma membrane with different lifetimes; some AP2 clusters are aborted rapidly (between 5–16 s) whereas others develop into fully committed pits (90 s) (**Loerke et al., 2009**). In the absence of FCHo, the lifetime of AP2 at the membrane in cultured cells is quite brief (<10 s), whereas

*Figure 5. Continued*

western blot analysis (anti-HA) 8 hr after heatshock (As in *Figure 3B*). Numbers indicate band intensity normalized to the fcho(+) sample. (**E**) Blot for phospho-rylated threonine 160. Samples collected before heatshock. Numbers indicate band intensity normalized to fcho(+) sample.

The following figure supplement is available for figure 5:

**Figure supplement 1**. An Inter-subunit salt bridge is broken in the active conformation of AP2.

overexpression of FCHo stabilizes AP2 and promotes the growth of clathrin-coated pits with long lifespans (>25 s) (*Henne et al., 2010*; *Cocucci et al., 2012*). We observe a similar phenotype in *fcho-1* mutants: although AP2 still associates with the plasma membrane, it does not form clusters. Moreover, the dwell time of AP2 on the membrane is shorter; in *fcho-1* mutants worms, the lifespan of AP2 on the membrane is reduced from 35 to 10 s. These results are strikingly similar to the observations of Cocucci et al (2012).

Previously it was thought that FCHo stabilizes AP2 on the membrane indirectly by binding Eps15 and intersectin, and these proteins in turn bind AP2 and stabilize the formation of a clathrin-coated pit (*Henne et al., 2010*). Our data suggest that FCHo acts directly on AP2: a 90 amino acid segment of the linker domain binds the AP2 complex in pulldowns from HEK293 cells. The linker region of human FCHo1 was previously shown to interact with AP2 (*Umasankar et al., 2012*). This interaction was thought to be via the appendage domain of the alpha subunit of AP2, however we find that the interior of the linker binds the core complex. This AP2 activator domain (APA) is conserved and this fragment from the nematode FCHO-1 protein or from the mammalian FCHo1, FCHo2 and SGIP1 proteins can rescue *fcho-1* mutant worms. It is curious to note that while the APA domain is conserved in metazoan orthologs of FCHO-1, it is absent in the yeast ortholog Syp1p (*Reider et al., 2009*).

What is the function of the µ-homology and F-BAR domains in *C. elegans*? Although the APA domain rescues *fcho-1* mutants to grossly wild-type morphology and behavior, it does not fully rescue at a cellular level: endocytosis of cargo is not restored to wild-type levels by expression of the APA fragment. Full rescue is only observed when the rescuing constructs include both the APA domain and the µ-homology domain. We have confirmed that the µ-homology domains of FCHo proteins bind Eps15 and Eps15-like proteins in pulldowns from HEK293 cells by extending the bait proteins to include this domain (*Figure 7—figure supplement 1C* and *Figure 7—figure supplement 2*). It is therefore likely that binding of FCHo to Eps15 is required for endocytosis of cargo. It has been demonstrated that FCHo forms an independent complex with Eps15 and Intersectin and that this complex functions in the recruitment of cargo to clathrin-coated pits (*Mayers et al., 2013*).

The F-BAR domain binds membrane and is required to recruit FCHo to the cell surface in both yeast and tissue culture cells (*Reider et al., 2009*; *Stimpson et al., 2009*; *Henne et al., 2010*). By contrast, neither the F-BAR domain of FCHO-1 or the membrane association domain of SGIP are required for rescue of *fcho-1* mutants. The dispensable nature of the F-BAR domain conflicts with models in which this domain must bend the membrane for clathrin-coated pit formation (*Henne et al., 2010*), and instead suggests that the most important feature of the F-BAR domain is its ability to localize the APA domain to the membrane. Apparently in *C. elegans*, the APA domain of FCHo can be recruited to the membrane via interactions with other proteins independent of the F-BAR domain. Nevertheless, the presence of a membrane-binding motif in all FCHo and SGIP proteins demonstrate that membrane interactions are important and conserved.

## Open AP2 bypasses the requirement for FCHo

Different crystal structures of AP2 suggest that the complex can adopt multiple conformations. AP2 can assume a closed and inert conformation in which membrane- and cargo-binding domains are inaccessible (*Collins et al., 2002*), in the unlatched or open conformations AP2 can bind the plasma membrane and the recognition motifs of cargo (*Kelly et al., 2008*; *Jackson et al., 2010*).

The gain-of-function mutations in AP2 that bypass the requirement for *fcho-1* provide in vivo support for these conformational changes. These mutations can be sorted into three classes based on the regions affected: the latch, the bowl and the hinge. In the unlatched state, the N-terminus of the beta subunit disconnects from the alpha and sigma2 subunits and exposes the dileucine-motif binding pocket (*Kelly et al., 2008*). Among the bypass suppressors of *fcho-1* were seven residues at the contact interface of the latched state. In the open structure, the mu domain is expelled from the bowl formed by the other subunits, and about half of the suppressors (34/71) were found in contact residues between mu2 and

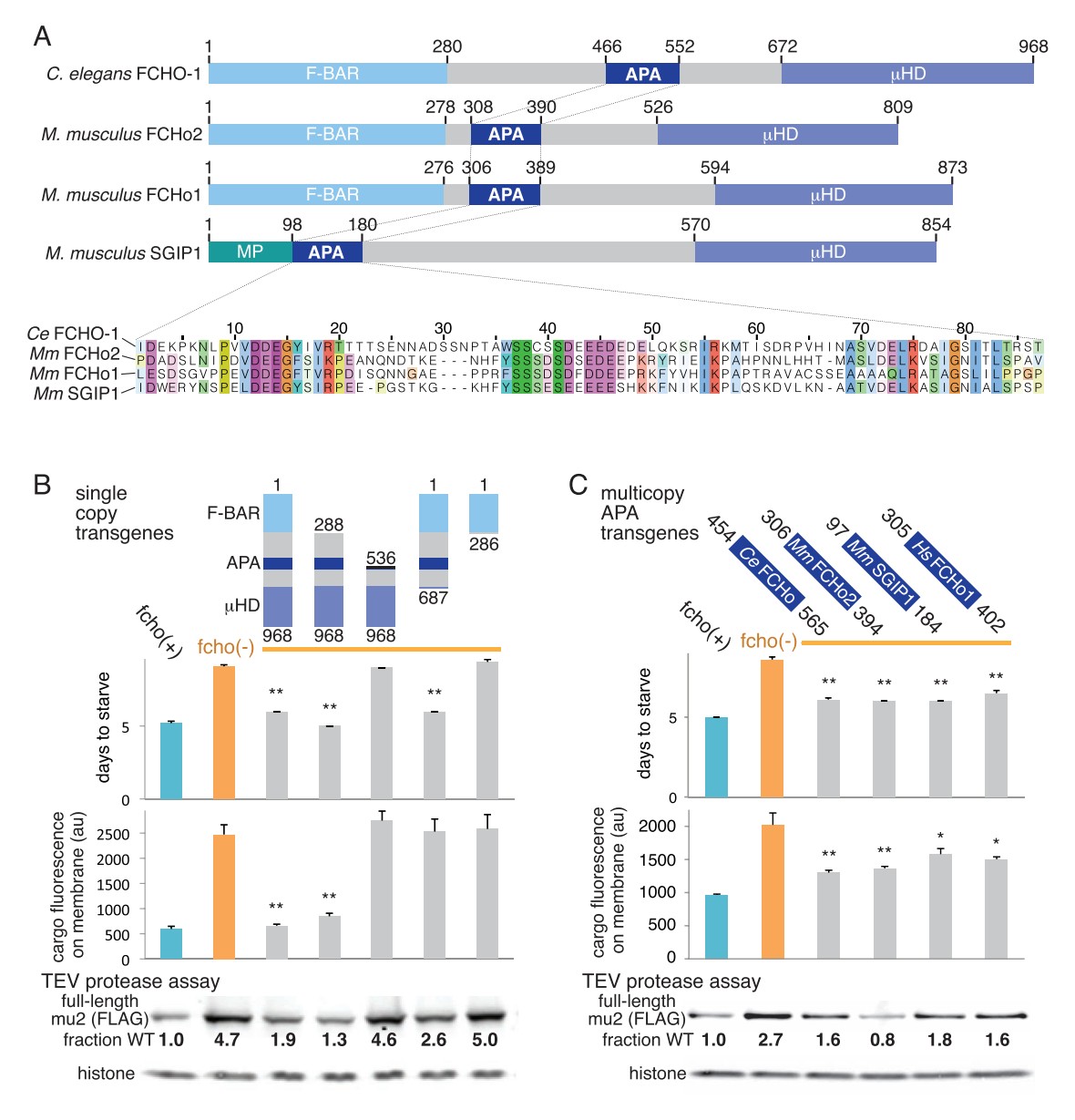

**Figure 6**. A Conserved region of FCHo proteins is necessary and sufficient to rescue *fcho-1* mutants. (**A**) FCHo homologs showing conserved domains. Amino acid numbers indicated above. The AP2 Activator (APA) domain is aligned below. Amino acids colored by Clustal X scheme and shaded by conservation. Membrane Phospholipid-binding domain (MP), μ-Homology Domain (μHD). (**B**) Structure/function analysis of worm FCHO-1. (**C**) Quantification of *fcho-1* mutant rescue with APA domains from worm (*Ce*), mouse (*Mm*), and human (*Hs*) orthologs expressed as extrachromosomal arrays. See *Figure 7—figure supplement 1A* for results of the starvation assay when the APA domains are expressed from single-copy transgenes. For (**B**) and (**C**), protease assay performed with FLAG-tagged mu2 subunit as in *Figure 3—figure supplement 1B*. Numbers indicate band intensity of full-length mu2 (top) relative to the histone control (bottom) and normalized to the fcho(+) sample.

The following figure supplement is available for figure 6:

**Figure supplement 1**. The APA domain of FCHO-1 Is sufficient to organize AP2 on the membrane.

the bowl. The alpha hinge domain flexes as the bowl collapses in the open state, and 19 mutations in the alpha hinge were identified. We isolated 7 other mutations in residues that reside near inter-subunit contacts in the closed conformation. These mutations are all consistent with a destabilization of the closed state, and are in fact in an open state as determined by the exposure of a TEV protease site in vivo.

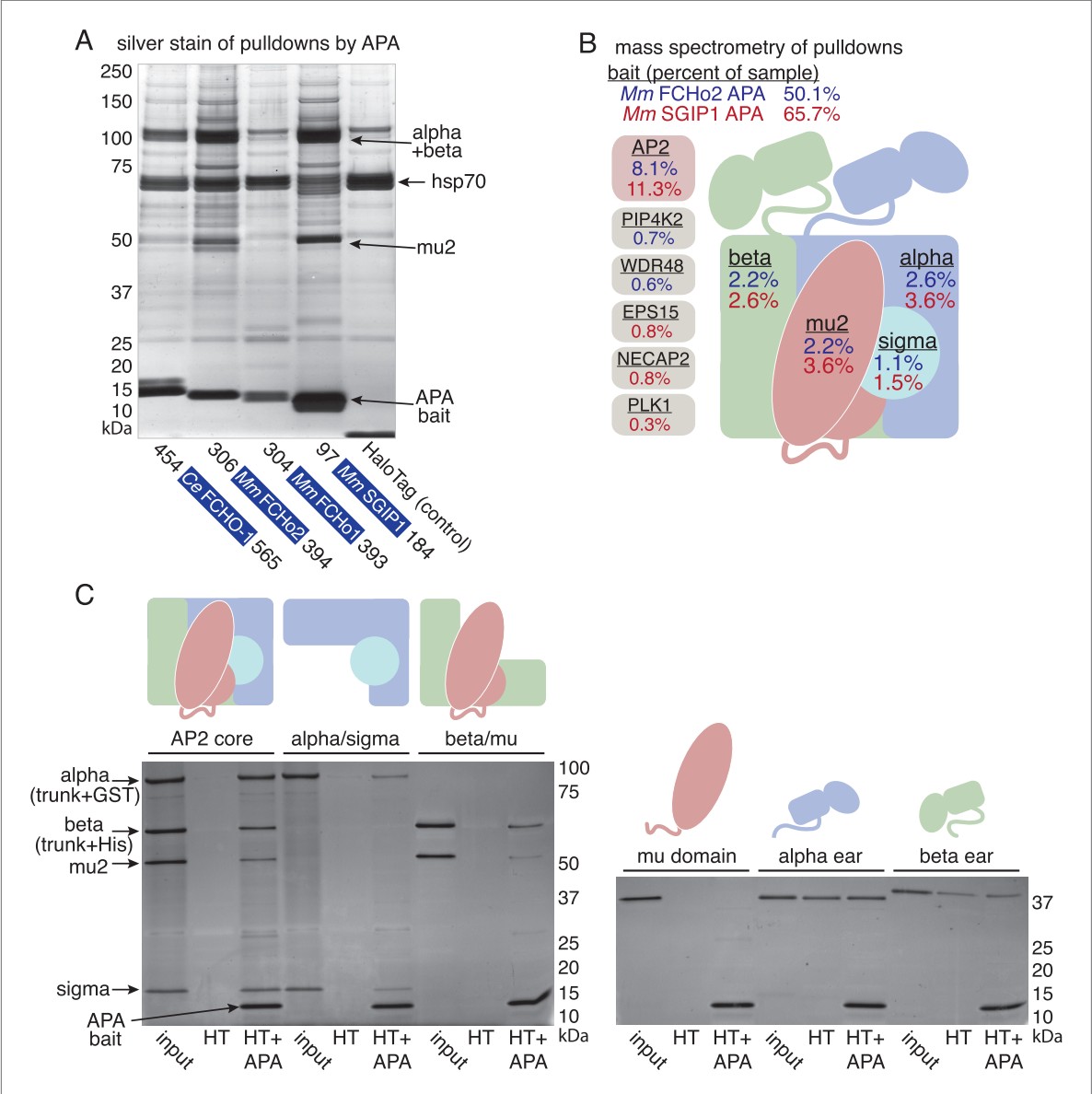

**Figure 7**. The APA domain binds AP2. In (**A**) and (**B**) APA domains from FCHo homologs were expressed as HaloTag fusions in HEK293T cells (*Ce*, *C. elegans*; *Mm*, *M. musculus*). (**A**) Silver-stained gel of affinity-purified proteins following proteolytic cleavage from the HaloTag. Arrows indicate bands of presumed identity. (**B**) The top ten human proteins purified using two different APA baits, as detected by MudPIT mass spectrometry. Nonspecific proteins also found in the control were removed. Values represent the mean % distributed Normalized Spectral Abundance Factor (dNSAF × 100) from three independent experiments. The values of all four AP2 subunits were totaled to determine the amount of complex in each sample. Multiple isoforms of alpha, beta, and phosphatidylinositol 5-phosphate 4-kinase type-2 (PIP4K2) were summed. WD repeat-containing protein 48 (WDR48), epidermal growth factor receptor substrate 15 (EPS15), adaptin ear-binding coat-associated protein 2 (NECAP2), and serine/threonine-protein kinase PLK1 (PLK1). See *Figure 7—figure supplement 2* for complete results. (**C**) APA pulldowns using bacterially expressed proteins. Purified HaloTag with (HT + APA) and without (HT) the APA domain from mouse SGIP1 were incubated with purified AP2 fragments followed by TEV protease cleavage to release the APA bait. Silver-stained gel of the eluted proteins. Note that the alpha/sigma and beta/mu hemicomplexes are soluble in our hands and that the AP2 appendage (ear) domains exhibit non-specific binding in this assay.

The following figure supplements are available for figure 7:

**Figure supplement 1**. The APA domain links FCHo proteins to the AP2 complex.

**Figure supplement 2**. Top proteins detected by MudPIT analysis.

The activated AP2 mutations that result in the most open (protease sensitive) and phosphorylated AP2 complex fully rescue the morphological and growth defects of the *fcho-1* deletion. Nevertheless, these mutations do not fully restore clearance of an artificial cargo from the surface of the intestine. It is likely that the activated AP2 mutations cannot recapitulate all of the normal functions of the AP2 complex, since it is known that FCHo has other functions beyond its actions on AP2, for example via interactions with Eps15 or Disabled-2 (*Figure 7—figure supplement 2* and *Figure 7—figure supplement 1C*) (*Reider et al., 2009*; *Henne et al., 2010*; *Uezu et al., 2011*; *Mulkearns and Cooper, 2012*; *Umasankar et al., 2012*; *Mayers et al., 2013*). Nor do these mutants exhibit enhanced endocytosis or membrane association in an otherwise wild-type background (*Figure 8*). It is possible that compensatory mechanisms counteract the open state of these AP2 mutants. Alternatively, our endocytosis assay may be at its detection limit because fluorescence from the artificial cargo is close to background levels in the wild-type.

The only mysterious suppressors are the mutations at the phosphorylation site on the mu2 subunit. The other suppressors led to an increase in phosphorylation of T160, consistent with phosphorylation promoting the open state (*Fingerhut et al., 2001*; *Olusanya et al., 2001*; *Ricotta et al., 2002*). However, mutation to a phosphodefective, as well as a phosphomimetic, amino acid caused AP2 to adopt the open state. These data are most consistent with dephospho-threonine at this position stabilizing the closed state. Phosphorylation then does not cause the open state but rather is a result of the open state. This conclusion is supported by the observation that clathrin assembly stimulates AAK1 to phosphorylate mu2 (*Conner et al., 2003*; *Jackson et al., 2003*).

How then does FCHo promote the AP2 cycle? The formation of a closed form of AP2 is probably required to unbind membranes from newly endocytosed vesicles and to scan the membrane for new sites of endocytosis. The coincidental presence of FCHo, cargo, and PIP2 can then stabilize the open state, and the conformational changes in the complex then nucleate recruitment of clathrin and other pit components (*Figure 4—figure supplement 1A*) (*Jackson et al., 2010*; *Kelly et al., 2014*).

## Materials and methods

### Strains

Worm strains were cultured and maintained using standard methods (*Brenner, 1974*). A complete list of strains and mutations used is in the Extended Strains List (*Supplementary file 1A*).

### Jowls screen

*C. elegans* late L4s were mutagenized for 4 hr at 22°C in 0.2 mM EMS. ~50,000 haploid genomes were screened in the $F_2$ and $F_3$ generation to isolate animals exhibiting the jowls phenotype. Genomic DNA was prepared from the offspring of these animals for amplification and subsequent sequencing of AP2 subunits. Mu2 subunit (*apm-2*) primer pairs were oGH408-9, oGH411-452, and oGH412-3. Alpha subunit (*apa-2*) primer pairs were oGH414-5, oGH416-7, and oGH418-9. Sigma2 subunit (*aps-2*) primer pair was oGH430-2. Beta subunit primer pairs were oGH441-2, oGH443-4, oGH445-6 and oGH447-8. To identify *fcho-1* mutants, PCR products corresponding to the coding sequences of the gene were amplified and sequenced using primer pairs oGH420-1, oGH423-50, oGH424-5, oGH428-9, and oGH433-51. Oligonucleotide sequences are listed in *Supplementary file 1C*.

### *fcho-1* alleles

The targeted deletion allele of *fcho-1* was generated by mobilizing a *Mos1* transposon from the gene (*ttTi3855*) and repairing the double strand break with a DNA template that replaces the first eight exons of *fcho-1* with an *unc-119*(+) transgene in an *unc-119(ed3)* mutant strain (*Frokjaer-Jensen et al., 2010*). The repair template plasmid was generated by Three-Fragment Multisite Gateway (Invitrogen, Carlsbad, CA). The proximal targeting arm (2.1 kb) was amplified (oligos oMT1-2, Mengyao Tan, University of Utah, Salt Lake City, UT) and cloned into the [2–3] donor. The distal arm (2.1 kb) was amplified with oMT3-4 and cloned into the [4–1] donor. The targeting arm entry clones were assembled with a [1–2] entry containing *unc-119(+)* (pRL8, Rachel Lofgren, University of Utah) in the [4–3] destination using *LR* clonase (Invitrogen). The resulting repair template was injected into *ttTi3855* II; *unc-119(ed3)* III worms along with additional plasmids (transposase and array markers) as previously described in (*Frokjaer-Jensen et al., 2010*) The molecular identity of the *dx34* allele was determined by whole genome sequencing (Illumina, San Diego, CA); *dx34* deletes the 3′ end of *fcho-1*, the downstream gene *vig-1* and the 5′ end of *jip-1*.

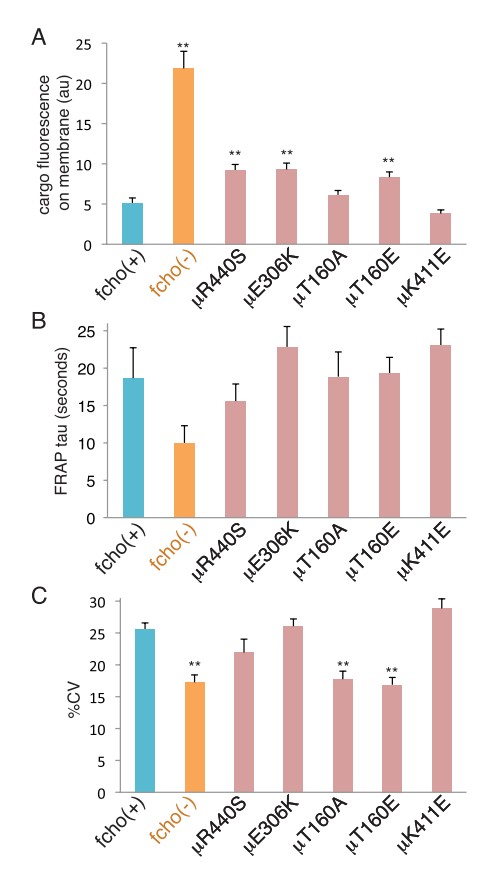

**Figure 8**. *fcho-1* bypass mutants do not exhibit hyperactive AP2. Mutations were examined in an fcho(+) background. (**A**) Cargo assay. Amount of GFP-tagged cargo on intestinal cell membrane. (**B**) FRAP assay. Time constants for recovery of alpha-GFP fluorescence after photobleaching. (**C**) The percent coefficient of variance of alpha-GFP pixel intensities in coelomocytes. Data represent mean ± SEM of n = 9–13 intestinal cells for (**A**) and n = 9–14 coelomocytes for (**B**) and (**C**); **p < 0.01 unpaired, two-tailed t-test compared to fcho(+).

Additional mutant alleles of *fcho-1 (ox500[Q634X], ox504 [frameshift], ox620[K872X], ox619[W882X])* were isolated in the jowls screen (See above). Worm strains, plasmids and oligonucleotide sequences are listed in *Supplementary file 1*.

## *fcho-1* suppressor screen

*fcho-1(ox477)* mutants were mutagenized for 4 hr at 22°C in 0.5 mM ENU. After washing with M9 buffer, ~100 L4 to young adult animals were pipetted onto 10 cm Normal Growth Media (NGM) agar plates previously seeded with 1 ml of a dense bacterial culture (OP50). After starvation, ~2 × 2 cm pieces of each plate were transferred to a fresh NGM agar plate with bacteria. This process was repeated 3–4 times to select for genotypes with greater fitness than the starting strain. ~10 worms were selected from each plate exhibiting a faster rate of food consumption than sibling plates. Genomic DNA was prepared from the offspring of these animals for amplification and sequencing of AP2 subunits. Males of *fcho-1* suppressor strains were generated by heatshock and crossed to *fcho-1* mutant hermaphrodites to score dominance and sex chromosomal linkage of the suppressor mutations in the $F_1$ offspring. *ox618* was isolated as a spontaneous suppressor. Worm strains are listed in *Supplementary file 1A*.

## Preparation of worms for microscopy

Worms were immobilized for fluorescence microscopy by placing them in a 1:1 mixture of a 1 µm polystyrene bead slurry (Polysciences, Warrington, PA) and 2× PBS pH 7.4 on 8–10% agarose pads (*Kim et al., 2013*). Worms were allowed to equilibrate on the slide for 5 min before data were acquired. Data acquisition from each slide did not last longer than 20 min to ensure the health of the worms.

## AP2 localization in coelomocytes

Worms expressing alpha adaptin-GFP (*oxSi254*) were imaged on an Ultraview VOX spinning disk confocal microscope (Perkin Elmer, Waltham, MA) with a 100× oil immersion objective (Carl Zeiss, Jena, Germany). A z-stack of half of the coelomocyte was acquired at maximum speed (200 ms exposure per slice) with 0.2 µm spacing. Fluorescence was excited with a 488 nm laser and was filtered through a 500–550 nm bandpass filter to an EMCCD (C9100-23B, Hamamatsu Photonics, Hamamatsu, Japan). The images were then analyzed using a set of custom written plugins (available at http://research.stowers.org/imagejplugins) in ImageJ (http://rsbweb.nih.gov/ij/). We started by creating a maximum intensity projection of the data. Next, the user specified an ROI inside the coelomocyte that does not include the outermost membrane. The mean intensity and standard deviation of the ROI was then measured for each time point. The coefficient of variance (%CV) was then calculated and averaged for all time points and all worms per sample. Additionally, histograms of pixel intensities inside the coelomocytes were taken from the exact same ROI as the %CV measurements. The mode of a logarithmic histogram with a bin size of 1000 was used as the background intensity. The mode was then subtracted from the histogram. A value of 15,000 was then added back to the image and a new logarithmic histogram with bin size of 1000 was measured.

This was necessary to avoid zero or negative values in the histogram. The bin width of the histogram was set to 1000 for all images, while the number of bins was allowed to vary. For each sample, all of the histograms were aligned to the mode and summed. Each point of the resulting histogram was then normalized to the integral of the histogram.

## Fluorescence recovery after photobleaching

FRAP data was recorded either on a Perkin Elmer Ultraview VOX spinning disk with a 63× 1.2 NA C-Apochromat water immersion objective or an LSM 780 confocal microscope with a 40× 1.1 NA C-Apochromat water immersion objective (Carl Zeiss). On both microscopes, fluorescence was excited with a 488 nm laser and emission was collected in the 500–550 nm range. On the LSM 780, emission was collected in photon counting mode range with a temporal resolution of 2 s and a pinhole close to 1 airy unit. Four images were acquired prior to bleaching of a manually selected ROI with maximum 488 nm laser power. On the spinning disk, emission was collected with an EMCCD camera (Hamamatsu C9100-23B). Five images were acquired at maximum speed (exposure 100–200ms) to determine the average fluorescence intensity of the coelomocyte membrane. Bleaching was then achieved over a user specified ROI, using a 488 nm laser (the duration of bleaching was less than five seconds). Recovery images were then captured for 3 min with a rate of 1 image per second. Data was analyzed using a custom written plugin (available at http://research.stowers.org/imagejplugins) for ImageJ (http://rsbweb.nih.gov/ij/). This macro first registered the image sequences to compensate for movement of the worm/coelomocyte. Then the user specified the bleach ROI. The mean intensity of the ROI was then plotted per time point. A fit to the data was achieved with a one component fluorescence recovery model. The tau (inverse rate) values from the fits were then averaged for all worms per sample. For average plots of fluorescence recovery, curves were normalized so that the minimum value was 0 and the maximum value was 1 prior to averaging.

## Cargo assay

Worms expressing the synthetic fluorescent cargo (GFP-CD4-YASV; *oxSi484*) were imaged on a LSM 780 confocal microscope (Carl Zeiss). A single cross-sectional Z-plane through the intestine was recorded. All of the images for a data set were recorded in a single session using the same laser settings. The images were analyzed in ImageJ. A segmented line was drawn on the basal-lateral membrane connecting intestinal segments 2 and 3, or rarely, 3 and 4. The average intensity along the line was recorded.

## Brood size assay

For each genotype, 10–12 L4 worms were singled to culture plates and transferred to a fresh plate every 12 hr. The transfers stopped when the worm burst (due to an egg-laying defect such as in AP2 mutants) or the worm started laying unfertilized oocytes (such as wild-type). The fertilized embryos from each animal were counted to determine the brood size. If the worm was lost during the transfer, the data were discarded.

## Starvation assay

NGM agar plates (60 mm) were seeded with 0.45 ml of bacterial culture (OP50 expanded overnight in 2xYT at 37°C without shaking). The bacterial lawns were grown at 22°C for 3 days. Three young adult hermaphrodites were placed on the bacteria and propagated at 22°C. Plates were examined daily until the worm population had consumed all of the bacteria and dispersed. Multiple (usually 10) plates were scored for each genotype.

## Molecular visualization

All structural representations in this paper were prepared with the PyMOL molecular graphics system, version 1.5.0.4 (Schrödinger, New York, NY; www.pymol.org). PyMOL visualization scripts are available at https://github.com/jorglab/Vu_AP2.

## TEV assay transgenes

A mini-gene encoding the mu2 subunit of AP2 (*apm-2*) was constructed using the Multisite Gateway System (Invitrogen). The *apm-2* cDNA was amplified by PCR using primers oGH634 and oGH635 and recombined with the [1–2] donor vector using BP clonase (Invitrogen) to generate the entry vector pGH442. The latter half of the cDNA was replaced with two genomic fragments corresponding to the last seven exons that were amplified with primer pairs oGH618-9 and oGH620-1. The PCR products were subsequently cloned into the *apm-2* cDNA entry vector amplified with oGH616-7 using the

Gibson assembly protocol (*Gibson et al., 2009*) to generate pGH443. A TEV protease cleavage site (ENLYFQGS) was inserted after the codon encoding alanine-240 using primers oGH756-7 and the Gibson reaction to generate pGH444. One version of the [1–2] *apm-2* entry vector was appended at the amino terminus with an HA tag (YPYDVPDYA) followed by a flexible linker (GTGGTGGSGGTG) by sequential amplification using primers oGH753-699 and oGH737-699 followed by recombination with the [1–2] donor to generate pGH445. A 3X FLAG tag (DYKDHDGDYKDHDIDYKDDDDK) was attached to a separate version of the [1–2] *apm-2* entry vector using primers oGH814-5 followed by the Gibson reaction to generate pGH446. 1.3 kb of the *apm-2* promoter region was amplified using oGH785-6 and recombined with the [4–1] donor vector to generate pGH461. The 3′ untranslated region (UTR) of *apm-2* was amplified with oGH797-519 and cloned into the [2–3] donor via the BP recombination reaction to generate pGH462. The entry clones were recombined with the [4–3] destination vector pCFJ606 (Christian Frøkjær-Jensen, University of Utah) using *LR* clonase (Invitrogen) to generate the complete *apm-2* minigene in a MosSCI targeting vector (*Frokjaer-Jensen et al., 2008*). The HA-tagged version is pGH447 and the Flag-tagged version is pGH448.

Mutations to the *apm-2* coding sequence of HA-tagged [1–2] *apm-2* minigene (pGH445) were introduced by PCR with primer pairs containing the mutation followed by the Gibson reaction to re-close the plasmid. The primer pairs and resulting [1–2] entry clones are oGH937-8 (pGH449) for E306K, oGH943-4 (pGH450) for K411E, oGH947-8 (pGH451) for R440S, oGH929-30 (pGH452) for T160A, and oGH931-2 (pGH453) for T160E. LR recombination with pGH461, pGH462, and pCFJ606, generated the MosSCI targeting vectors pGH454 for E306K, pGH455 for K411E, pGH456 for R440S, pGH457 for T160A, and pGH458 for T160E.

To generate an inducible TEV protease, a codon-optimized protease sequence containing two artificial introns was synthesized as two gBlocks (IDT, Coralville, IA) and assembled with the *hsp-16.41* promoter and *unc-54* 3′UTR to generate pGH459 by digesting pWD141 (M Wayne Davis, University of Utah) with BstBI and EcoRV to generate a vector backbone for the Gibson reaction. The subsequent *Phsp:TEV:unc-54UTR* sequence was amplified (oGH806-7) and inserted into pCFJ150 amplified with oGH751-2 using Gibson assembly to generate a MosSCI targeting vector called pGH460. See *Supplementary file 1* for oligonucleotide sequences, plasmids, and worm strains.

## FCHo structure/function transgenes

The full-length *fcho-1* cDNA (2.9 kb) was amplified (oGH323-4) and recombined with the [1–2] donor vector of the Multisite Gateway (Invitrogen) three-fragment system using BP clonase (Invitrogen). A tagRFP containing three artificial introns (Stefan Eimer, University of Freiburg, Freiburg im Breisgau, Germany), an engineered S158T mutation (Rob Hobson, University of Utah) and flanked by flexible linkers (N-term: STSGGSGGTGGS; C-term: GGTGGTGGSGGTG) was amplified (oCF590-1) and inserted after the start codon of the *fcho-1* cDNA using oGH350-352 to open the vector and the Gibson reaction to close it. An HA tag was inserted into the N-terminal linker of TagRFP by digestion with KpnI followed by ligation with two annealed oligos encoding the tag (oGH372-3). The resulting [1–2] entry clone encoding HA_TagRFP_wormFCHO-1(1-968) was pGH389. Deletions of *fcho-1* coding sequence corresponding to amino acids 1-287 (F-BAR), 1-535 (F-BAR + APA), 688-968 (µHD), and 287-968 (APA + µHD) were introduced to the [1–2] entry using PCR and Gibson assembly. The resulting plasmids were: pGH475 for FCHO-1(288-968), pGH388 for FCHO-1(536-968), pGH476 for FCHO-1(1-687), and pGH477 for FCHO-1(1-286).

For APA expression, sequences corresponding to the same regions used as bait in tissue culture cells (See previous section) were amplified (oGH793-4 for *C. elegans* FCHO-1, oGH808-9 for *Mus musculus* FCHo2, oGH810-1 for *M. musculus* FCHo1, oGH812-3 for *M. musculus* SGIP1, and oGH1035-6 for *Homo sapiens* FCHo1) and inserted after the C-terminal linker of TagRFP using vector primers oGH649 and oGH781 with Gibson assembly. The resulting [1–2] entry clones were: pGH478 for worm FCHO-1(454-565), pGH479 for mouse FCHo2(306-394), pGH480 for mouse FCHo1(304-393), pGH481 for mouse SGIP1(97-184), and pGH482 for human FCHo1(305-402). All [1–2] entry clones were recombined with a [4–1] entry containing the ubiquitous *dpy-30* promoter, the *unc-54* 3′UTR in a [2–3] entry and one of two [4–3] destination vectors (pCFJ201 or pCFJ212) using LR clonase (Invitrogen) to generate MosSCI targeting vectors (*Frokjaer-Jensen et al., 2008*). The resulting plasmids were: pGH394 for FCHO-1(1-968), pGH483 for FCHO-1(288-968), pGH393 for FCHO-1(536-968), pGH484 for FCHO-1(1-687), pGH485 for FCHO-1(1-286) pGH486 for FCHO-1(454-565), pGH487 for mouse FCHo2(306-394), pGH488 for mouse FCHo1(304-393), pGH489 for mouse SGIP1(97-184), and pGH490 for human

FCHo1(305-402). See *Supplementary file 1* for oligonucleotide sequences, plasmids, and worm strains.

## Heatshock and western blot analysis

Heatshock was performed by sealing worm plates with Parafilm and submerging them in a 34°C circulating water bath for 1 hr. For each sample, 100 or 200 L4 stage animals were selected and placed in microfuge tubes containing M9 buffer +0.001% Triton X-100. The worms were washed once with M9 buffer +0.001% Triton X-100 and collected by centrifugation (1000×g, 30 s) and placed on ice. All but ~10 µl of the buffer was removed and 10 µl of LDS Sample buffer (4×, Novex, Invitrogen) with ~100 mM fresh dithiothreitol was added. Samples were frozen in liquid N2 and stored at −80°C. Samples were then sonicated at 0°C for 6 min at 100% amplitude in a cup horn (Branson) and denatured at 99°C for 5 min. Entire lysates were loaded into NuPage 4–12% Bis-Tris Gels (Novex) for electrophoresis followed by transfer to nitrocellulose membranes using the iBlot system (Novex). For anti-HA blots, membranes were blocked in Tris Buffered Saline with 0.1% Tween 20 (TBST) and 5% milk powder. Anti-HA-Peroxidase High Affinity (3F10, Roche, Indianapolis, IN) was diluted 1:200 in TBST with 1% milk powder. Peroxidase was detected with ChemiGlow (Protein Simple, San Jose, CA) and imaged on a G:BOX (Syngene). For all other antigens, blocking and antibody incubations occurred in Odyssey Blocking Buffer (LI-COR, Lincoln, NE). Primary antibodies and dilutions include mouse anti-FLAG (1:1000, M2, Sigma-Aldrich, St. Louis, MO), rabbit anti-APM1 (phospho T156, 1:1000, Abcam 109397, Cambridge, England), rabbit anti-histone H3 (1:100000, Abcam 1791), and rabbit-anti TEV protease (1:500, Rockland Immunochemicals, Limerick, PA). Fluorescent secondary antibodies include goat anti-mouse IRDye 680LT (1:20,000, LI-COR) and goat anti-rabbit IRDye 800CW (1:15,000, LI-COR). All washes were performed in TBST. Band intensities were quantified using ImageStudioLite (LI-COR).

## Affinity purification of proteins from tissue culture cells

For preparation of samples for MudPIT analysis, sequences corresponding to the APA domains of FCHo proteins were inserted following the HaloTag (Promega, Madison, WI) sequence in a modified version of pcDNA5/frt (*Banks et al., 2014*). The plasmid was linearized with PacI/PmeI and assembled using the Gibson reaction with each APA domain amplified from cDNA. The corresponding amino acids, primers and plasmids were 454–565 of *C. elegans* FCHO-1 (NM_061546.3; oGH828-9; pGH463), 306-394 of *M. musculus* FCHo2 (NM_172591.3; oGH830-1; pGH464), 304-393 of *M. musculus* FCHo1 (NM_028715.3; oGH832-3; pGH465), and 97-184 of *M. musculus* SGIP1 (AB262964.1; oGH834-5; pGH466). To identify proteins interacting with additional regions of FCHo proteins, amino acids 1-809 (oGH886-7; pGH467) and 263-809 (oGH887-892; pGH468) of *M. musculus* FCHo2 and 1-854 (oGH890-1; pGH469) of *M. musculus* SGIP1 were cloned. For western blot analysis of FCHo1 interactions, additional sequences encoding amino acids 267-609 of *M. musculus* FCHo1 (oGH1019-20; pGH470), 305-402 (oGH1039-40; pGH471) and 267-609 (oGH1021-2; pGH472) of *H. sapiens* FCHo1 (NM_001161357.1), and 295-390 (oGH1041-2; pGH473) and 260-609 (oGH1023-4; pGH474) of *Danio rerio* FCHo1 (XM_005166937.1) were also cloned into the same mammalian expression vector. Plasmids and oligonucleotides are listed in *Supplementary files 1B,C*, respectively. 150 mm dishes of HEK293T cells (~80% confluent, Tissue Culture Core, Stowers Institute for Medical Research, Kansas City, MO) were transfected with 10 µg of plasmid using Lipofectamine 2000 (Invitrogen). 24–36 hr later, cells were washed with PBS, scraped from the dishes, collected by centrifugation and frozen at −80°C. Cell pellets were lysed and bound to HaloLink Magnetic Beads (Promega) according to the manufacturer's instructions. After washing the beads, complexes were released by incubation (2–3 hr at 22°C while shaking) with AcTEV protease (2 units in 100 µl, Invitrogen) to digest the cleavage site between the HaloTag and bait proteins. 20 µl of samples destined for MudPIT analysis were separated by electrophoresis and visualized using the Silver Stain Plus Kit (Bio-Rad, Hercules, CA). The remaining 80 µl were precipitated using trichloroacetic acid. For western blot analysis of FCHo1 interactors, 20 µl of purified complexes were electrophoresed, transferred, and blotted as described above using the following primary antibodies: mouse anti-alpha adaptin (610501; 1:2000; BD Biosciences, San Jose, CA), rabbit anti-AP2B1 (151961; 1:1000; Abcam), rabbit anti AP2M1 (75995; 1:1000; Abcam) and rabbit anti-AP2S1 (128950; 1:10000; Abcam).

## MudPIT analysis

TCA-precipitated proteins were urea-denatured, reduced, alkylated and digested with endoproteinase Lys-C (Roche) followed by modified trypsin (Promega) (*Washburn et al., 2001*; *Florens and Washburn, 2006*). Peptide mixtures were loaded onto 250 µm fused silica microcapillary columns

packed with strong cation exchange resin (Luna, Phenomenex, Torrance, CA) and 5-µm $C_{18}$ reverse phase (Aqua, Phenomenex), and then connected to a 100 µm fused silica microcapillary column packed with 5-µm $C_{18}$ reverse phase (Aqua, Phenomenex) (*Florens and Washburn, 2006*). Loaded microcapillary columns were placed in-line with a Quaternary Agilent 1100 series HPLC pump and a LTQ linear ion trap mass spectrometer equipped with a nano-LC electrospray ionization source (ThermoScientific, San Jose, CA). Fully automated 10-step MudPIT runs were carried out on the electrosprayed peptides, as described in (*Florens and Washburn, 2006*). Tandem mass (MS/MS) spectra were interpreted using SEQUEST (*Eng et al., 1994*) against a database consisting of 30,499 non-redundant human proteins (NCBI, 2012-08-27 release), 160 usual contaminants (human keratins, IgGs, and proteolytic enzymes), as well the mouse and *C. elagans* FCHo constructs and the mouse SGIP1 sequences. To estimate false discovery rates (FDR)s, the amino acid sequence of each non-redundant protein entry was randomized to generate a virtual library. This resulted in a total library of 61,327 non-redundant sequences against which the spectra were matched. Peptide/spectrum matches were sorted and selected using DTASelect (*Tabb et al., 2002*) with the following criteria set: Spectra/peptide matches were only retained if they had a DeltCn of at least 0.08, and minimum XCorr of 1.8 for singly-, 2.0 for doubly-, and 3.0 for triply-charged spectra. In addition, peptides had to be fully tryptic and at least seven amino acids long. Combining all runs, proteins had to be detected by at least two such peptides, or one peptide with two spectra. Under these criteria the averaged FDRs at the protein and peptide levels were 0.24% ± 0.2 and 0.44% ± 0.3, respectively. Peptide hits from multiple runs were compared using CONTRAST (*Tabb et al., 2002*). To estimate relative protein levels, distributed Normalized Spectral Abundance Factors (dNSAFs) were calculated for each detected protein/protein group, as described in (*Zhang et al., 2010*). The open source BioConductor package plgem in R was used to statistically compare the proteins detected in the FCHo and SGIP1 samples to negative controls (*Pavelka et al., 2008*). Proteins were considered significantly enriched compared to the control datasets if their p-values for power law global error model signal-to-noise (PLGEM-STN) ratios were lower than 0.001, and they were detected in at least 2 out of 3 replicate analyses of the FCHo and SGIP1 purifications. The top 10 proteins (in addition to bait) were ranked based on decreasing PLGEM-STN values for FCHo2 APA, decreasing STN and dNSAF values for SGIP1 APA and full length (FL) SGIP1, and decreasing dNSAF values for FCHo2 FL and FCHo2 without the F-BAR domain (ΔBAR).

## Recombinant protein purification and pulldown

The His-tagged beta-mu hemicomplex expression vector is pGH424: Sequence corresponding to trunk domain of mouse AP2 beta 1 (amino acids 1–591, NM_001035854) was amplified from mouse brain cDNA (Elena Gracheva, University of California, San Francisco, CA) with primers oGH368 and oGH676; the fragment was inserted into the pETduet-1 vector amplified using oGH332 and oGH336 using the Gibson assembly reaction. This places the coding sequence downstream of the first T7 promoter. The C-terminus is tagged with a three amino acid linker (GSS) followed by a hexa histidine-tag (His-tag). The coding sequence of mouse AP2 mu1 (amino acids 1–435, NM_009679) was amplified (oGH370 + oGH371) and inserted downstream of the second T7 promoter (oGH338 + oGH339) using the Gibson reaction. The His-tagged mu domain of AP2 mu1 expression construct is pGH441: This construct was generated by amplifying a portion of pGH424 (oGH571 + oGH921) starting with the linker and mu domain of the mu2 protein, continuing around the plasmid backbone, and ending with the first T7 promoter. This PCR product removed the beta trunk along with the sigma-homology domain of mu while appending a His-tag to the N-terminus of the linker, and was circularized using the Gibson assembly reaction. The His-tagged beta appendage domain expression construct is pGH491: Sequence encoding the C-terminus of mouse AP2 beta 1 (amino acids 592–951) was amplified using oGH1161-2 and recombined with the His-tag, T7 promoter and vector backbone portion of the His-tagged mu domain construct (pGH441 amplified with pGH571 and oGH339) using Gibson assembly. The His-tagged alpha appendage expression construct is pGH492: cDNA corresponding to the linker + appendage domain (amino acids 622–938) of mouse AP2 alpha 2 (NM_007459) was amplified (oGH1163-4) and cloned using the same strategy as the beta appendage expression construct (pGH491, above). The His-tagged HaloTag-APA expression vector is pGH493: The His-tag, T7 promoter and backbone regions of the mu domain construct (pGH441 amplified with pGH853 and oGH1165) was recombined with PCR products corresponding to HaloTag (amplified using oGH1166-7) and the APA domain of mouse SGIP1 (oGH1169 + oGH861) using the Gibson reaction. The control construct to express the His-tagged HaloTag alone in bacteria is pGH494: The HaloTag coding sequence was amplified (oGH1166 + oGH1171) and recombined with the His-tag, T7 promoter and

backbone regions of the AP2 mu domain construct (pGH441 amplified with oGH1170 + oGH1165). The vector expressing the trunk domain of mouse AP2 alpha 2 with a C-terminal GST tag along with mouse AP2 sigma 2 (*Collins et al., 2002*) was a gift from Volker Haucke (FMP, Berlin, Germany). Plasmids and oligonucleotides are listed in *Supplementary files 1B,C*, respectively.

Each AP2 hemicomplex was expressed independently. Expression vectors were transformed into Rosetta (DE3)pLysS cells (EMD Millipore, Billerica, MA) and grown overnight at 20°C in LB containing 200 µM IPTG, chloramphenicol (34 µg/ml), and ampicillin (100 µg/ml) for the His-tagged proteins and kanamycin (25 µg/ml) for the alpha/sigma hemicomplex. Cells were collected by centrifugation, washed with distilled water and re-pelleted prior to rapid freezing. Cell pellets were resuspended in lysis buffer (50 mM HEPES, 300 mM NaCl, 10 mM imidazole, pH 7.5) and incubated with 1 mg/ml lysozyme at 4°C prior to sonication and centrifugal clarification. To purify His-tagged proteins, lysates were incubated with Talon cobalt resin (Clontech, Mountain View, CA), washed with lysis buffer containing 20 mM imidazole and eluted with lysis buffer containing 200 mM imidazole. To purify the alpha/sigma hemicomplex, lysate was incubated with glutathione agarose (Pierce, Thermo Fisher Scientific, Rockford, IL) and then washed with lysis buffer prior to eluting with lysis buffer containing 10 mM reduced glutathione. Purified proteins were dialyzed in 25 mM HEPES, 100 mM KCl, pH 7.5 and stored as frozen aliquots at −80°C. For the pulldown assay, 80 pmoles of purified HaloTag ± APA bait were diluted along with 40 pmoles of recombinant AP2 prey and 10 µl of magnetic HaloLink beads (20% slurry) in 1x TBS containing 0.05% IGEPAL CA-630 (1 ml total volume for each pulldown) and nutated overnight at 4°C. Beads were washed with 1× TBS +0.05% IGEPAL CA-630 and bound proteins were cleaved from the HaloTag by incubation with AcTEV protease (30 µl at 50 units/ml for 60 min at 22°C). 50% of this elution was separated by SDS-PAGE and silver-stained along with 25% of the prey input for comparison. AP2 hemicomplexes are soluble under these conditions. To purify AP2 for crystallographic studies it was found that hemicomplexes were insoluble (*Collins et al., 2002*). Note that the protein concentrations used in our pulldown assay (~10 µg/ml) are roughly 1000-fold lower than those indicated for the crystallization (~10 mg/ml).

## Acknowledgements

We thank Alejandro Sánchez Alvarado and the Stowers Institute for Medical Research for support. Ho Yi Mak for providing lab space, reagents, and technical advice. Tatjana Piotrowski for the use of the LSM780 confocal microscope. Kausik Si for proposing to identify APA interactors in HEK293 cells. Charles Banks for technical advice on biochemistry and reagents for purifying APA interactors from HEK293 cells. Mahadevan Lakshminarasimhan for guidance on recombinant protein expression and purification. Nadja Makki and Kim Schuske for mapping the *fcho-1(dx34)* allele. Mengyao Tan for constructing the *fcho-1* MosDEL targeting vector. M Wayne Davis and Christian Frøkjær-Jensen for suggestions to improve the manuscript. This work was supported by NIH NS034307 and NSF IOS-0920069 to EMJ.

## Additional information

### Funding

| Funder | Grant reference number | Author |
| --- | --- | --- |
| Howard Hughes Medical Institute | | Gunther Hollopeter, Thien N Vu, Mingyu Gu, Michael Ailion, Erik M Jorgensen |
| Stowers Institute for Medical Research | | Gunther Hollopeter, Jeffrey J Lange, Ying Zhang, Brian D Slaughter, Jay R Unruh, Laurence Florens |
| National Institute of Neurological Disorders and Stroke | NS034307 | Erik M Jorgensen |
| Division of Integrative Organismal Systems | IOS-0920069 | Erik M Jorgensen |

The funders had no role in study design, data collection and interpretation, or the decision to submit the work for publication.

## Author contributions

GH, Conception and design, Acquisition of data, Analysis and interpretation of data, Drafting or revising the article; JJL, Microscopy, Acquisition of data, Analysis and interpretation of data; YZ, Mass spectrometry, Acquisition of data, Analysis and interpretation of data; TNV, Molecular visualization, Analysis and interpretation of data; MG, Built the artificial cargo, Contributed unpublished essential data or reagents; MA, Genetics, Analysis and interpretation of data; EJL, Isolated the *fcho-1(dx34)* allele, Contributed unpublished essential data or reagents; BDS, Microscopy, Acquisition of data; JRU, Microscopy, Analysis and interpretation of data; LF, Mass spectrometry, Analysis and interpretation of data; EMJ, Conception and design, Analysis and interpretation of data, Drafting or revising the article

## Additional files

### Supplementary file

• Supplementary file 1. Worm strains, plasmids, and oligonucleotide sequences. (**A**) Extended Strains List (**B**) Plasmids (**C**) Oligonucleotides.

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
