## [Decision Letter]

Thank you for sending your work entitled “FCHo and SGIP proteins are allosteric activators of the AP2 clathrin adaptor complex” for consideration at *eLife.* Your article has been favorably evaluated by Vivek Malhotra (Senior editor) and 3 reviewers, one of whom, Suzanne Pfeffer, is a member of our Board of Reviewing Editors.

This is an elegant study from the Jorgensen lab that demonstrates a role for the endocytic adaptor FCHo in promoting AP-2 function. Taking advantage of *C. elegans* genetics, the authors identify an enormous set of mutations within subunits of the AP-2 complex during an *fcho* mutant suppressor screen. The mutations cluster in a manner that suggests they stabilize an open conformation for the AP-2 complex. Indeed, using an elegant in vivo protease sensitivity assay, several bypass mutants show enhanced cleavage, suggesting that they exist in a more open state. Additional experiments, including a clever second site suppressor analysis, further support this conclusion. The authors also identify a short, conserved sequence in FCHo (the APA domain), which is sufficient to promote AP-2 activity and rescue a subset of phenotypes exhibited by *fcho* mutant animals. In general, the work is rigorous and convincing, finally highlighting the relevance of different conformational states to AP-2 function in vivo. Nonetheless, the following issues should be addressed before publication can be recommended.

1) The significance of the paper would be enhanced if the authors would map the interaction between the proteins more precisely and provide an estimate of the affinity of the interaction; they should also provide specific information in terms of how the fragment affects AP2 behavior on the membrane.

2) A predicted AP-2 cargo is shown to accumulate on the surface of intestinal cells when FCHo is absent. However, there a pool of the cargo that continues to accumulate intracellularly in this mutant background. How does the *fcho* mutant phenotype compare to that of a strain that lacks the alpha subunit of AP-2? And in the double mutant lacking both the alpha subunit and *fcho*? The issue here is really trying to distinguish whether FCHo has functions outside of its role as an AP-2 activator. Since none of the bypass mutants could fully suppress the *fcho* deletion, it would suggest that FCHo functions independently of AP-2 as well. Can the authors clarify (either with this intestinal endocytosis assay or otherwise)?

3) Is there a consequence (with regard to endocytosis) to animals expressing a constitutively active form of AP-2 (e.g., the bypass mutants)? Is endocytosis enhanced in animals that harbor bypass mutations in an otherwise wild type background? If not, can the authors comment on why a transition from a closed to an open state might be beneficial?

4) In a similar vein, is the mobility/clustering of the alpha subunit altered in animals expressing one of the bypass mutants, as measured by FRAP? One might expect that the constitutively open state may translate into enhanced stabilization at the cell surface. Such a finding would be highly informative and enhance the impact of this study.

5) Continuing on this line of reasoning, what is the effect of expressing the APA domain on the mobility/clustering of the alpha subunit in coelomocytes?

6) The rescue of *fcho* mutants by the APA domain alone is somewhat surprising, as it is unclear how this fragment would be appropriately targeted. Perhaps its interaction with AP-2 is sufficiently strong to enable its localization? Unfortunately, there is no evidence to support this possibility at present. Can the authors express the domain (perhaps as a fluorescently tagged protein) and determine whether it accumulates at the cell surface, as one would anticipate? In this way, the authors could also determine which subunit (or at least hemicomplex) of AP-2 is necessary for targeting, as localization studies could be conducted in both alpha and mu deletion backgrounds.

---

## [Author Response]

*1) The significance of the paper would be enhanced if the authors would map the interaction between the proteins more precisely and provide an estimate of the affinity of the interaction; they should also provide specific information in terms of how the fragment affects AP2 behavior on the membrane*.

The reviewers suggested that the interaction of the APA domains of FCHo and SGIP with AP2 be further characterized: We determined that the APA domain binds specifically and directly to both hemicomplexes of the AP2 core. We expressed His-tagged or GST-tagged AP2 subunits (or domains) in bacteria, and separately expressed a Halo-tagged SGIP1 APA fragment and purified the recombinant proteins. The Halo-tagged APA fragment was conjugated to beads; the AP2 constructs were affinity-purified on these beads and identified on silver-stained gels. These data are included in Figure 7.

Results section: “We demonstrated that the interaction between the APA domain and the AP2 core is direct using purified recombinant proteins in pulldown assays (Figure 7). The APA domain does not appear to bind the appendages of the large adaptins, nor to the mu domain alone. Rather, it bridges the complex since the APA bait binds both the alpha/sigma and beta/mu hemicomplexes.”

Discussion section: “… we find that the interior of the linker binds the core complex.”

We attempted to determine the affinity of the APA domain for the AP2 core using Fluorescent Correlation Spectroscopy. We labeled the purified HaloTag proteins with the fluorescent ligand TMR and incubated with increasing concentrations of the purified AP2 core complex. We repeated these experiments three times but the results were not significant. This is partly due to the fact that when the fluorescent Halo-tagged APA binds to the AP2 core, the shift in diffusion rate is not large enough to consistently rise above the baseline nor the diffusion rate of our negative control (purified HaloTag lacking the APA). We do see a deflection in the diffusion rate of the fluorescent APA domain when the concentration of AP2 core approaches 1 μM, however we are not convinced and do not feel comfortable publishing these data.

When overexpressed in *fcho-1* mutants, the APA domain alone is sufficient to increase AP2 association with the membrane as assayed by FRAP. Moreover, the APA fragment can organize the complex into puncta on the coelomocyte membrane and is thus likely assembling into presumptive endocytic pits. These data are shown in Figure 6—figure supplement 1.

In the text: “Expression of this short fragment alone was capable of rescuing *fcho-1* mutants, including growth rate, and endocytosis of cargo (Figure 6 and Figure 7—figure supplement 1). This fragment is also sufficient to immobilize AP2 on the membrane in the photobleaching assay and to cluster AP2 into presumptive endocytic pits (Figure 6—figure supplement 1).”

*2) A predicted AP-2 cargo is shown to accumulate on the surface of intestinal cells when FCHo is absent. However, there a pool of the cargo that continues to accumulate intracellularly in this mutant background. How does the* fcho *mutant phenotype compare to that of a strain that lacks the alpha subunit of AP-2? And in the double mutant lacking both the alpha subunit and* fcho*? The issue here is really trying to distinguish whether FCHo has functions outside of its role as an AP-2 activator. Since none of the bypass mutants could fully suppress the* fcho *deletion, it would suggest that FCHo functions independently of AP-2 as well. Can the authors clarify (either with this intestinal endocytosis assay or otherwise)?*

The reviewers are asking whether FCHo has functions beyond activation of AP2. We have now tested this by assays in double knockouts in the pathway, in *fcho-1* mutants expressing just the APA fragment, and in *fcho-1* null mutants with suppressors (as described below). Although we lack data demonstrating other roles for FCHo, we think that it is quite likely given the conservation of the mu homology domain and its known interactions. We describe these new data in the Results and now acknowledge that FCHo is likely to have other roles in the Discussion.

The first and best method of testing for other roles is to assay for enhanced phenotypes in mutants lacking both FCHo and AP2. We found that surface retention of the artificial cargo was not enhanced in the double mutants lacking *fcho-1* and each of the AP2 subunits alpha, sigma, or mu.

In the text: “However, this cargo accumulates on the cell surface in both AP2 and *fcho-1* mutants, suggesting that AP2-dependent endocytosis is defective in *fcho-1* mutants (Figure 1—figure supplement 1). In addition the phenotype is not enhanced in double mutants indicating that FCHO-1 acts in the same pathway as AP2.”

The second method is to assay *fcho-1* nulls expressing the APA activating fragment. The APA fragment almost fully rescues *fcho-1* phenotypes for morphology, growth and endocytosis.

In the text: “Expression of this short fragment alone was capable of rescuing *fcho-1* mutants, including growth rate, endocytosis of cargo, and morphology (Figure 6, Figure 7—figure supplement 1 and not shown).”

The third method is to assay *fcho-1* nulls with suppressor mutations. The activated AP2 mutations that result in the most open (protease sensitive) and phosphorylated AP2 complex fully rescue the morphological and growth defects of the *fcho-1* deletion (Figure 4).

Nevertheless, these mutations do not fully restore clearance of an artificial cargo from the surface of the intestine. There are two possible reasons: either the activated AP2 mutations cannot recapitulate all of the normal functions of the AP2 complex, or there are other functions of FCHo beyond its actions on AP2.

In the text: “However, none of these mutations fully restored cargo endocytosis, even though all of the suppressors rescued growth and morphology (Figure 4 and not shown). Only the mutation that resulted in a profoundly protease-hypersensitive complex (μE306K) increased cargo internalization with high significance in *fcho-1* mutants. These findings indicate that subtle conformational changes favoring active AP2 satisfied an organismal requirement for FCHO-1 without fully compensating for the endocytic defect of *fcho-1* mutants.”

Discussion section: “The activated AP2 mutations that result in the most open (protease sensitive) and phosphorylated AP2 complex fully rescue the morphological and growth defects of the *fcho-1* deletion. Nevertheless, these mutations do not fully suppress clearance of an artificial cargo from the surface of the intestine. It is likely that the activated AP2 mutations cannot recapitulate all of the normal functions of the AP2 complex, and that FCHo has other functions beyond its actions on AP2, for example via interactions with Eps15 or Disabled-2 (Figure 7—figure supplement 2 and Figure 7—figure supplement 1) (16; 26; 27; 35; 43; 44).”

3) Is there a consequence (with regard to endocytosis) to animals expressing a constitutively active form of AP-2 (e.g., the bypass mutants)? Is endocytosis enhanced in animals that harbor bypass mutations in an otherwise wild type background? If not, can the authors comment on why a transition from a closed to an open state might be beneficial?

The reviewers are asking whether open AP2 mutants exhibit enhanced endocytosis in the intestine. We performed the cargo assay in five strains bearing activated mu2 mutations in an otherwise wild-type background (Figure 8). None of these mutants internalize more of the artificial cargo than the wild type. We have not observed an overt phenotype associated with constitutively open AP2 other than their ability to bypass the requirement for FCHo/SGIP proteins.

In the text: “Nor do these mutants exhibit enhanced endocytosis or membrane association in an otherwise wild-type background (Figure 8). It is possible that compensatory mechanisms counteract the open state of these AP2 mutants.”

Why is a transition from a closed to open state likely to be beneficial? The existence of a closed, soluble form of AP2 is probably important for trafficking the complex from recently formed vesicles back to the plasma membrane to initiate new sites of endocytosis. The open form is configured to bind membrane and cargo simultaneously, and the change in conformation in the superstructure probably signals the recruitment of clathrin.

If the constitutively activated forms cannot adopt the closed state, why don’t they affect endocytic trafficking? A complex that is in the open state could both enhance and slow cycling of the complex and may therefore superficially resemble the wild type, or the opening of AP2 may not be rate-limiting, or additional mechanisms may prevent AP2 hyperactivity.

In the text: “How then does FCHo promote the AP2 cycle? The formation of a closed form of AP2 is probably required to unbind membranes from newly endocytosed vesicles and to scan the membrane for new sites of endocytosis. The coincidental presence of FCHo, cargo, and PIP2 can then stabilize the open state, and the conformational changes in the complex then nucleate recruitment of clathrin and other pit components (Figure 4—figure supplement 1; [20]).”

*4) In a similar vein, is the mobility/clustering of the alpha subunit altered in animals expressing one of the bypass mutants, as measured by FRAP? One might expect that the constitutively open state may translate into enhanced stabilization at the cell surface. Such a finding would be highly informative and enhance the impact of this study*.

The activating mutations do not appear to significantly stabilize or cluster the AP2 complex on the cell surface beyond levels exhibited by the wild-type AP2 complex. We analyzed the same five bypass mutants evaluated in the cargo assay above using FRAP to monitor the mobility of AP2 (Figure 8) and variance analysis to quantify the clustering of AP2 (Figure 8). None were significantly better than the wild-type AP2 complex. Again, it is possible that constitutively open AP2 elicits compensatory mechanisms or that the mutations are not so severe as to lead to a constitutively adherent complex.

Same quote as for point 3: “Nor do these mutants exhibit enhanced endocytosis or membrane association in an otherwise wild-type background (Figure 8). It is possible that compensatory mechanisms counteract the open state of these AP2 mutants.”

5) Continuing on this line of reasoning, what is the effect of expressing the APA domain on the mobility/clustering of the alpha subunit in coelomocytes?

Overexpression of the APA domain rescues the mobility and the clustering of the alpha subunit in coelomocytes (see Figure 6—figure supplement 1).

In the text: “Expression of this short fragment alone was capable of rescuing *fcho-1* mutants, including growth rate, endocytosis of cargo, and morphology (Figure 6, Figure 7—figure supplement 1, and not shown). This fragment is also sufficient to immobilize AP2 on the membrane in the photobleaching assay and to cluster AP2 into presumptive endocytic pits (Figure 6—figure supplement 1).”

*6) The rescue of* fcho *mutants by the APA domain alone is somewhat surprising, as it is unclear how this fragment would be appropriately targeted. Perhaps its interaction with AP-2 is sufficiently strong to enable its localization? Unfortunately, there is no evidence to support this possibility at present. Can the authors express the domain (perhaps as a fluorescently tagged protein) and determine whether it accumulates at the cell surface, as one would anticipate? In this way, the authors could also determine which subunit (or at least hemicomplex) of AP-2 is necessary for targeting, as localization studies could be conducted in both alpha and mu deletion backgrounds*.

All of the structure-function FCHO-1 constructs, and mammalian FCHo and SGIP constructs were tagged at the N-terminus with TagRFP-T (Figure 6). We examined the localization of the APA domain from mouse SGIP in coelomocytes (Figure 7—figure supplement 1). The RFP-APA accumulates on the cell surface and overlaps with the GFP-tagged AP2 alpha subunit. In the absence of the mu subunit, the APA fragment is no longer enriched on the cell surface, even though the AP2 alpha-GFP retains the ability to localize to the plasma membrane (although not in a productive manner). This result demonstrates that the AP2 holocomplex is required to generate the APA binding site and is consistent with our *in vitro* biochemistry.

In the text: “The interaction of the APA domain with AP2 likely occurs *in vivo* as well, since fluorescently tagged APA colocalizes with AP2 on the membrane in coelomocytes, and membrane association is lost in mutants lacking the mu2 subunit (Figure 7—figure supplement 1).”